# Mechanism Design as Inverse Games: A Computational Approach to Dynamic Mechanism Design

## Abstract

Mechanism design is often described as inverse game theory: rather than analyzing equilibria of a game, the designer specifies rules to induce desirable outcome at equilibrium. We present a computational framework for optimal dynamic mechanism design with evolving agent types. We cast the problem as a constrained optimization over partially observable Markov games, with incentive compatibility and individual rationality encoded as constraints. To solve it, we develop a min–max optimization approach and propose two methods for handling partial observability: (i) Bayesian belief-state tracking with convergence guarantees in discrete-type settings, and (ii) recurrent neural embeddings that scale to continuous types. In bandit auction experiments, our framework recovers known single-item benchmarks and discovers new incentive-compatible mechanisms in multi-item environments lacking analytical solutions.

## 1 Introduction

*Mechanism design* is often described as *inverse game theory*: rather than predicting equilibrium outcomes in a given game, the principal (i.e., the designer) begins with a desired outcome and works backwards to design rules of interaction so that strategic behavior by self-interested agents yields the desired outcome (i.e., equilibrium). *Dynamic mechanism design (DMD)* extends this paradigm to environments where decisions unfold over multiple periods and agents' types (i.e., their private information) evolve over time. This problem is pivotal in many real-world applications, such as auctions for multi-period goods (e.g., spectrum licenses), long-term contracting in supply chains, and subscription-based services. The added comlexity of temporal and informational dynamics makes DMD especially challenging, and explicit mechanism characterizations are rare.

An important strand of the dynamic mechanism design literature studies profit-maximizing mechanisms in dynamic settings with evolving private information (Courty & Hao, 2000; Battaglini, 2005; Eso & Szentes, 2007; Kakade et al., 2013). In particular, Pavan et al. (2014) synthesize earlier work and  develop a comprehensive dynamic contracting framework that accommodates arbitrary horizons, multiple agents, a continuum of types, and serial correlation with dependence on past allocations—serving as the dynamic counterpart to the Myersonian static framework (Myerson, 1981). Despite the generality of the framework, successful mechanism characterization is only achieved for restrictive settings (e.g., dynamics follows a specific form and, most importantly, types are unidimensional) — a limitation shared by analytical approaches.

In this paper, we approach DMD from the inverse-game perspective.  An *inverse game* contains a parameterized game in which the agents' equilibrium behavior is known (via data or analytics), and the goal is to find parameters of the game that induce the observed behavior. Goktas et al. (2024) employed representation learning within this perspective: they represent parameterized games as expressive neural networks and search the induced game space for parameters that rationalize the given equilibrium. A solution to this inverse game is called an *inverse equilibrium* (Goktas et al., 2024). An insight that makes this inverse-game perspective promising for DMD is: while computing the mapping from a game to its equilibria is PPAD-hard even in simple normal-form games (Daskalakis et al., 2009), computing the inverse mapping—from observed behavior to its inverse equilibrium—is polynomial-time tractable for a large class of games (Goktas et al., 2024).

Applying this perspective to DMD, we represent mechanisms using expressive neural networks and establish a mapping from the network's parameter space, to the parameterized mechanism space, and the space of corresponding induced games. We then explore the mechanism-induced game space, searching for parameters that rationalize any desired outcome i.e., inverse equilibria. In this framework, such inverse equilibria are mechanisms.

By the dynamic revelation principle (Sugaya & Wolitzky, 2021), without loss of generality, we restrict our attention to *direct* dynamic mechanisms that are *incentive compatible (IC)*, i.e., truthful reporting constitutes an equilibrium in the induced game. Together with *individual rationality (IR)* (i.e., voluntary participation constraints), these conditions define the target space of mechanisms. That is, it suffices to search the space of mechanism-induced games for parameters that rationalize truthful reporting—i.e., to search for an *inverse truthful equilibrium*—and incentivize agents to participate. We extend this perspective to *optimal* mechanism design by nesting the aforementioned constraints of the mechanism-induced game within a bilevel optimization, i.e., we search for an inverse truthful equilibrium that optimizes the principal's objective.

This inverse-game perspective resonates with the paradigm of *automated mechanism design (AMD)* (Conitzer & Sandholm, 2002; 2004; Zhang & Conitzer, 2021), especially recent work in the area of *differentiable economics (DE)* (Dütting et al., 2023). DE can be reinterpreted through this lens: by optimizing mechanisms subject to zero-regret incentive-compatibility constraints, DE implicitly enforces truthful reporting as an equilibrium of an induced game, i.e., DE is a search through the space of inverse truthful equilibria for an optimal mechanism. DE has been applied to unidimensional static mechanism design problems, recovering known analytical solutions, and to multidimensional static mechanism design problems, where analytical methods break down. Similarly, our work recovers known analytical solutions to unidimensional *dynamic* mechanism design problems, and immediately generalizes to multidimensional *dynamic* mechanism design problems.

**Contribution.** This paper introduces a computational framework for optimal dynamic mechanism design. Our approach builds on the intuition that mechanism design is inherently inverse game theory, and leverages modern differentiable tools to explore the mechanism-induced game space directly.

In Section 3, we develop a general model of DMD, extending the framework in Pavan et al. (2014). Specifically, we utilize partially observable Markov games (POMGs) as the foundational game model, as it captures the partial observability and dynamic nature of the problem. We then formulate optimal DMD as an optimization problem over the induced game space. In Section 4, we propose a solution procedure with two approaches for handling partial observability: 1. explicit Bayesian belief updates, which reduce POMGs to Markov games and yield polynomial-time convergence to IC+IR mechanisms—locally optimal in discrete-type compressed-information settings and globally optimal in contextual-bandit settings and 2. recurrent neural embeddings of private information, which bypass explicit belief tracking and scale to continuous types. In Section 5, we evaluate our approaches in bandit auctions: in discrete-type and continuous-type single-item auctions, belief-state tracking and RNN-based private information embeddings find IC+IR mechanisms that achieve the same expected revenue as analytical benchmarks (Pavan et al., 2014) respectively, while in multi-item settings, where closed-form solutions are not known, our methods discover mechanisms that satisfy IC and IR constraints and achieve high payoff for the principal.

## 2 PRELIMINARIES

For readability, detailed notation and definitions are deferred to Appendix B.

**Parameterized Partially Observable Markov Games.** Given a parameter space $\Theta$ with parameter $\boldsymbol{\theta} \in \Theta \subseteq \mathbb{R}^d$, a *parameterized partially observable Markov game (POMG)* is a tuple $\mathcal{Y}^{\boldsymbol{\theta}} \doteq (n, T, \mathcal{S}, \mathcal{A}, P, \gamma, \mu, r, \mathcal{O}, O, \boldsymbol{\theta}, \Theta)$, within the space of parameterized POMGs $\mathcal{Z}^{\Theta}$, s.t.:

- Horizon $T$: A positive integer or $\infty$.
- State space $\mathcal{S}$: A nonempty Borel space.
- Action spaces $\{\mathcal{A}_i\}_{i \in [n]}$: Each $\mathcal{A}_i$ is a nonempty Borel space, and we denote the space of joint actions by $\mathcal{A} = \bigtimes_{i \in [n]} \mathcal{A}_i$.
- Parameterized transition kernel $P : \mathcal{S} \times \mathcal{S} \times \mathcal{A} \times \Theta \to [0, 1]$: $P$ is a Borel-measurable stochastic kernel on $\mathcal{S}$ given $\mathcal{S} \times \mathcal{A} \times \Theta$.
- Discount factor $\gamma$.

- Initial state distribution $\mu \in \Delta(\mathcal{S})$: A probability measure on $\mathcal{S}$.
- Parameterized reward functions $\{r_i : \mathcal{S} \times \mathcal{A} \times \Theta \to \mathbb{R}\}_{i \in [n]}$: Each $r_i$ is a Borel-measurable function from $\mathcal{S} \times \mathcal{A} \times \Theta$ to $\mathbb{R}$.
- Observation spaces $\{\mathcal{O}_i\}_{i \in [n]}$: Each $\mathcal{O}_i$ is a nonempty Borel space, and we denote the space of joint observations by $\mathcal{O} = \bigtimes_{i \in [n]} \mathcal{O}_i$.
- Observation kernels $\{O_i : \mathcal{O}_i \times \mathcal{S} \to [0,1]\}_{i \in [n]}$: Each Borel-measurable $O_i$ is a stochastic kernel on $\mathcal{O}_i$ given $\mathcal{S}$,[1] with the joint observation kernel $O : \mathcal{O} \times \mathcal{S} \to [0,1]$.

The game initiates at time $\tau = 0$ in some state $s_0 \sim \mu$ drawn from an initial state distribution $\mu$. At each time period $\tau \in [(T-1)^*]$, each player $i \in [n]$ first receives *observation* $o_{i,\tau}$ which is stochastically generated via the observation kernel $O_i(do_{i,\tau} \mid s_\tau)$, plays an *action* $a_{i,\tau} \in \mathcal{A}_i$, and receives a *reward* $r_i(s_\tau, \boldsymbol{a}_\tau; \boldsymbol{\theta})$. The game then transitions to a new state $s_{\tau+1}$, following probability distribution $P(ds^{\tau+1} \mid s^\tau, \boldsymbol{a}_\tau; \boldsymbol{\theta})$.

Define $\mathcal{I}_{i,\tau} = (\mathcal{O}_i \times \mathcal{A}_i)^\tau \times \mathcal{O}_i$ as the *$\tau$th information space for player $i$*, with $\iota_{i,\tau} = ((o_{i,k}, a_{i,k})_{k=0}^{\tau-1}, o_{i,\tau}) \in \mathcal{I}_{i,\tau}$ as the *$\tau$th information vector for player $i$*. Likewise, define $\mathcal{H}_\tau = (\mathcal{S} \times \mathcal{O} \times \mathcal{A})^{\tau+1}$ as the *$\tau$th history space*, and $\boldsymbol{h}_\tau = (s_k, \boldsymbol{o}_k, \boldsymbol{a}_k)_{k=0}^\tau \in \mathcal{H}_\tau$ as then *$\tau$th history vector*.

A *policy* for player $i$ is a sequence $\boldsymbol{\pi}_i = (\pi_{i,0}, \pi_{i,1}, \cdots, \pi_{i,T-1})$ such that for each $\tau \in [T^*]$, $\pi_{i,\tau}$ is a universally measurable stochastic kernel on $\mathcal{A}_i$ given $\mathcal{I}_{i,\tau}$. If, for each $\iota_{i,\tau} \in \mathcal{I}_{i,\tau}$, $\pi_{i,\tau}(da_{i,\tau} \mid \iota_{i,\tau})$ assigns mass one to some point in $\mathcal{A}_i$, $\boldsymbol{\pi}_i$ is *deterministic*. In this case, by a slight abuse of notation, $\boldsymbol{\pi}_i$ can be considered a sequence of universally measurable mappings $\pi_{i,\tau} : \mathcal{I}_{i,\tau} \to \mathcal{A}_i$. We refer to the space of all *deterministic* policies for player $i$ as $\mathcal{P}_i^{\mathrm{PO}}$. As usual, $\boldsymbol{\pi} \doteq (\boldsymbol{\pi}_1, \ldots, \boldsymbol{\pi}_n) \in \mathcal{P}^{\mathrm{PO}} \doteq \bigtimes_{i \in [n]} \mathcal{P}_i^{\mathrm{PO}}$ denotes a deterministic policy profile.

Given a policy profile $\boldsymbol{\pi} \in \mathcal{P}$ and an initial state distribution $\mu \in \Delta(\mathcal{S}_0)$, we denote the *$\tau$th-step history distribution measure* on $\mathcal{H}_\tau$ by $\nu_\mu^{\boldsymbol{\pi}, \boldsymbol{\theta}, \tau}$ (see Appendix B for formal definition). Given a policy profile $\boldsymbol{\pi} \in \mathcal{P}$, player $i$'s *payoff* $U_i(\boldsymbol{\pi}; \boldsymbol{\theta}) \doteq \mathbb{E}_{H \sim \nu_\mu^{\boldsymbol{\pi}, \boldsymbol{\theta}, T-1}} \left[ \sum_{\tau=0}^{T-1} \gamma^\tau r_i(S_\tau, A_\tau; \boldsymbol{\theta}) \right]$. For all $\boldsymbol{\pi}, \boldsymbol{\pi}' \in \mathcal{P}^{\mathrm{PO}}$, the *cumulative regret* of $\boldsymbol{\pi}$ relative to $\boldsymbol{\pi}'$ is $\Psi(\boldsymbol{\pi}, \boldsymbol{\pi}'; \boldsymbol{\theta}) \doteq \sum_{i \in [n]} U_i(\boldsymbol{\pi}_i', \boldsymbol{\pi}_{-i}; \boldsymbol{\theta}) - U_i(\boldsymbol{\pi}; \boldsymbol{\theta})$. Moreover, given a deterministic policy profile $\boldsymbol{\pi} \in \mathcal{P}^{\mathrm{PO}}$, the *exploitability* of $\boldsymbol{\pi}$ is $\varphi(\boldsymbol{\pi}; \boldsymbol{\theta}) \doteq \max_{\boldsymbol{\pi}' \in \mathcal{P}} \Psi(\boldsymbol{\pi}, \boldsymbol{\pi}'; \boldsymbol{\theta})$, which represents the sum of the players' maximal unilateral payoff deviations. As usual, an *$\varepsilon$-Bayesian Nash equilibrium* ($\varepsilon$-BNE) of a POMG $\mathcal{Y}^{\boldsymbol{\theta}}$ is a policy profile $\boldsymbol{\pi}^* \in \mathcal{P}^{\mathrm{PO}}$ such that for all players $i \in [n]$, $U_i(\boldsymbol{\pi}^*; \boldsymbol{\theta}) \geq \max_{\boldsymbol{\pi}_i \in \mathcal{P}_i} U_i(\boldsymbol{\pi}_i, \boldsymbol{\pi}_{-i}^*; \boldsymbol{\theta}) - \varepsilon$. In particular, a Bayesian Nash equilibrium is realized when $\varepsilon = 0$.

## 3 OPTIMAL DYNAMIC MECHANISM DESIGN

**Dynamic Mechanism Design** A *dynamic mechanism design (DMD)* problem $\mathcal{D} \doteq (n, T, \mathcal{T}, \mathcal{X}, \boldsymbol{\omega}, F, \boldsymbol{u}, u_0, \gamma)$ comprises a principal and $n$ agents. At each time period $\tau \in [T^*]$, each agent receives a private type $t_{i,\tau} \in \mathcal{T}_i$ (when $\tau = 0$, $t_{i,0}$ is drawn from the initial type distribution $\omega_i \in \Delta(\mathcal{T}_i)$, with product measure $\boldsymbol{\omega} = \bigotimes_{i \in i} \omega_i$), sends report $\hat{t}_{i,\tau} \in \mathcal{T}_i$ to the principal, and receives an outcome $x_{i,\tau} \in \mathcal{X}_i$, which is publicly observed by all agents[2] [3]. Agent $i$ will then receive *immediate reward* $u_i(t_{i,\tau}, \boldsymbol{x}^\tau)$ given her current type and all past joint outcomes, and principal will receive $u_0(\boldsymbol{t}_\tau, \boldsymbol{x}^\tau)$ given current type profile and past joint outcomes. Then, each agent $i$'s type $t_{i,\tau}$ evolves to $t_{i,\tau+1}$, according to the probability distribution $F_i(t_{i,\tau+1} \mid t_{i,\tau}, \boldsymbol{x}^\tau)$.[4] We denote the joint type evolution kernels by $F : \mathcal{T} \times \mathcal{T} \times \bigcup_{\tau=1}^{T} \mathcal{X}^\tau \to [0,1]$.

---

[1]More generally, each observation kernel is a stochastic kernel on $\mathcal{O}_i$ given $\mathcal{A} \times \mathcal{S}$, but we simplify it here to keep the model concise.

[2]This assumption is consistent with the literature (Bergemann & Välimäki, 2019).

[3]For convenience, for all $i \in [n]$ and $0 \leq \tau \leq T$, let $x_i^\tau = (x_{i,k})_{k=0}^\tau$, $t_i^\tau = (t_{i,k})_{k=0}^\tau$, and $\hat{t}_i^\tau = (\hat{t}_{i,k})_{k=0}^\tau$ denote the collection of agent $i$'s past outcomes, past types, and past reports from time period 0 to $\tau$, respectively. An analogous notation extends joint outcome, type profile, and report profile.

[4]We assume the environment satisfies the *Markov property*: each agent's reward $u_{i,\tau}$ depends only on her current type $t_{i,\tau}$ rather than all past types $t_i^\tau$; principal's reward is similar; the type transition kernel $F_i$ depends only on the current type $t_{i,\tau}$ rather than all past types $t_i^\tau$. This assumption is not strictly necessary, as one could always augment the state space with type histories, but it enables a more tractable computational analysis.

Given a DMD $\mathcal{D}$, a *dynamic (direct) mechanism* is an *outcome rule* $g : \bigcup_{\tau=1}^{T} \mathcal{T}^{\tau} \to \mathcal{X}$ that maps past report profiles $\hat{\boldsymbol{t}}_{\tau} \in \mathcal{T}^{\tau+1}$ of any length $\tau + 1$ to a joint outcome $\boldsymbol{x}_{\tau} = g(\hat{\boldsymbol{t}}^{\tau})$. We denote the space of *dynamic (direct) mechanisms* by $\mathcal{G}$.

For all agents $i \in [n]$ participating in $\mathcal{D}$, a *reporting strategy* is a collection $\{\pi_{i,\tau} : \mathcal{T}_i^{\tau+1} \times \mathcal{T}_i^{\tau} \times \mathcal{X}^{\tau} \to \mathcal{T}_i\}_{\tau=0}^{T-1}$, where each $\pi_{i,\tau}(t_i^{\tau}, \hat{t}_i^{\tau-1}, \boldsymbol{x}^{\tau-1})$ is agent $i$'s report at time $\tau$ when her current and past true types are $t_i^{\tau}$, her reported past types are $\hat{t}_i^{\tau-1}$, and the joint past outcomes are $\boldsymbol{x}^{\tau-1}$. Truthtelling is the reporting strategy that always reports true types, i.e., $\pi_{i,\tau}(t_i^{\tau}, \hat{t}_i^{\tau-1}, \boldsymbol{x}^{\tau-1}) = t_{i,\tau}$, for all $\tau \in [(T-1)^*]$.

A solution to a DMD problem is a mechanism that is (i) *incentive compatible (IC)*, meaning each agent maximizes cumulative reward by truthfully reporting, and (ii) *individually rational (IR)*, meaning each agent has incentive to participate. Thus, this problem can naturally viewed as a *sequential decision-making problem* for the principal, with global IC and IR constraints. The state-of-the-art DMD literature (Pavan et al., 2014) takes exactly this perspective, but the difficulty lies in reducing global IC constraints to recursive (i.e., local) IC constraints and encoding them back into the optimization. While this difficulty can be resolved in unidimensional-type settings, all known techniques break down in multi-dimensional type spaces.

To tackle multi-dimensional DMD problems, we follow the paradigm of inverse game theory (Goktas et al., 2024). That is, we parameterize dynamic mechanisms using expressive neural network representations, and then establish a mapping between the network's parameter space $\Theta$, the parameterized dynamic mechanism space $\mathcal{G}^{\Theta}$, and the space of their induced games $\mathcal{Z}^{\Theta}$. This mapping enables us to directly explore the induced game space, searching for parameters for which truthful reporting is an equilibrium and voluntary participation is incentivized, by-passing the need for recursive characterizations as shown in Figure 1 .

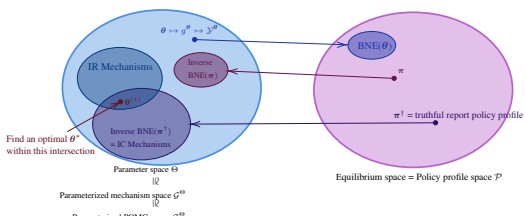

Figure 1: Illustration of mapping between spaces.

**Inverse Game-Theoretic Formulation of Dynamic Mechanism Design**      Consider a class of *parameterized dynamic mechanisms* $\mathcal{G}^{\Theta}$ for a DMD problem $\mathcal{D}$. For each parameter $\boldsymbol{\theta} \in \Theta$, $g^{\boldsymbol{\theta}} \in \mathcal{G}^{\Theta}$ is a parameterized (direct) dynamic mechanism, which induces an *Agents POMG* $\mathcal{Y}^{\boldsymbol{\theta}}$, where

- The state space $\mathcal{S} = \bigcup_{\tau=0}^{T-1} \mathcal{T} \times \mathcal{X}^{\tau} \times \mathcal{T}^{\tau}$: each $s_{\tau} = (\boldsymbol{t}_{\tau}, \boldsymbol{x}^{\tau-1}, \hat{\boldsymbol{t}}^{\tau-1}) \in \mathcal{S}$ encodes the current type profile $\boldsymbol{t}_{\tau}$, past joint outcomes $\boldsymbol{x}^{\tau-1}$, and past report profiles $\hat{\boldsymbol{t}}^{\tau-1}$.
- The action spaces $\{\mathcal{A}_i = \mathcal{T}_i\}_{i \in [n]}$ are type spaces, and the agents' actions are their reports.
- The parameterized transition kernel $P : \mathcal{S} \times \mathcal{S} \times \mathcal{A} \times \Theta \to [0, 1]$ is defined as $P(s_{\tau+1} \mid s_{\tau}, \boldsymbol{a}_{\tau}; \boldsymbol{\theta}) = \mathbb{1}_{\{\boldsymbol{x}_{\tau}\}}(g^{\boldsymbol{\theta}}(\hat{\boldsymbol{t}}^{\tau-1}, \boldsymbol{a}_{\tau})) \mathbb{1}_{\{\hat{\boldsymbol{t}}^{\tau}\}}(\hat{\boldsymbol{t}}^{\tau-1}, \boldsymbol{a}_{\tau}) F(\boldsymbol{t}_{\tau+1} \mid \boldsymbol{t}_{\tau}, \boldsymbol{x}_{\tau})$, for all $s_{\tau} = (\boldsymbol{t}_{\tau}, \boldsymbol{x}^{\tau-1}, \hat{\boldsymbol{t}}_{\tau-1})$ and $s_{\tau+1} = (\boldsymbol{t}_{\tau+1}, \boldsymbol{x}^{\tau}, \hat{\boldsymbol{t}}^{\tau})$.
- The discount factor $\gamma$ is inherited from $\mathcal{D}$.
- The initial state distribution $\mu \in \Delta(\mathcal{S})$ is defined as $\mu(s_0) = \boldsymbol{\omega}(\boldsymbol{t}_0)$, for all $s_0 = (\boldsymbol{t}_0)$.
- The parameterized reward functions $r_i : \mathcal{S} \times \mathcal{A} \times \Theta \to \mathbb{R}$ are defined as $r_i(s_{\tau}, \boldsymbol{a}_{\tau}; \boldsymbol{\theta}) = u_i(t_{i,\tau}, (\boldsymbol{x}^{\tau-1}, g^{\boldsymbol{\theta}}(\hat{\boldsymbol{t}}^{\tau-1}, \boldsymbol{a}_{\tau})))$, for all $s_{\tau} = (\boldsymbol{t}_{\tau}, \boldsymbol{x}^{\tau-1}, \hat{\boldsymbol{t}}^{\tau-1})$.
- The observation spaces $\{\mathcal{O}_i = \mathcal{X} \times \mathcal{T}_i\}_{i \in [n]}$ are the Cartesian products of the joint outcome and individual type spaces. At each time step $\tau \in T$, agent $i$ observes the last joint outcome and her current true type, i.e., $o_{i,\tau} = (\boldsymbol{x}_{\tau-1}, t_{i,\tau})$.
- The observation kernels $\{O_i : \mathcal{O}_i \times \mathcal{S} \to [0, 1]\}_{i \in [n]}$ are defined as $o_{i,\tau} = (\boldsymbol{x}, t_i)$, $O_{i,\tau}(o_{i,\tau+1} \mid s_{\tau}) = \mathbb{1}_{\{\boldsymbol{x}\}}(\boldsymbol{x}_{\tau-1}) \mathbb{1}_{\{t_i\}}(t_{i,\tau})$, for all $s_{\tau} = (\boldsymbol{t}_{\tau}, \boldsymbol{x}^{\tau-1}, \hat{\boldsymbol{t}}^{\tau-1})$.

Note that in this formulation, the $\tau$th information space for player $i$, $\mathcal{I}_{i,\tau} = (\mathcal{O}_i \times \mathcal{A}_i)^{\tau} \times \mathcal{O}_i = \mathcal{T}_i^{\tau+1} \times \mathcal{T}_i^{\tau} \times \mathcal{X}^{\tau}$, is the space of her past and current true types, her past reports, and past joint outcomes. Therefore, $\boldsymbol{\pi}_i = (\pi_{i,0}, \cdots, \pi_{i,T-1})$ is a deterministic policy, where each $\pi_{i,\tau} : \mathcal{I}_{i,\tau} \to \mathcal{A}_i$ is a measurable mapping, corresponding to a reporting strategy for $i$.

We can now formally define the DMD incentive compatibility and individual rationality conditions. Consider a DMD problem $\mathcal{D}$, a parameterized (direct) dynamic mechanism $g^{\boldsymbol{\theta}}$, and the associated *Agents POMG* $\mathcal{Y}^{\boldsymbol{\theta}}$. Let $\boldsymbol{\pi}^{\dagger}$ be the policy profile corresponding to the truthtelling strategy profile. We say that $g^{\boldsymbol{\theta}}$ is (i) *Bayesian incentive compatible (BIC)* if $\boldsymbol{\pi}^{\dagger}$ is a Bayesian-Nash equilibrium of $\mathcal{Y}^{\boldsymbol{\theta}}$ and (ii) *Bayesian individually rational (BIR)* if $U_i(\boldsymbol{\pi}^{\dagger}; \boldsymbol{\theta}) \geq 0$, for all agents $i \in [n]$.

**Optimal Dynamic Mechanism Design** Having formally defined BIC and BIR, we can now formalize optimal DMD: to identify a BIC, BIR dynamic mechanism that maximizes the principal's expected payoff. In this section, we formulate this task as an optimization problem over the mechanism-induced parameterized game space—in particular, as a search for an optimal inverse truthful equilibrium.

Consider a DMD problem $\mathcal{D}$, a parameterized (direct) dynamic mechanism $g^{\boldsymbol{\theta}}$, and the associated *Agents POMG* $\mathcal{Y}^{\boldsymbol{\theta}}$. Based on $\mathcal{Y}^{\boldsymbol{\theta}}$, we can further define the principal's parameterized reward functions $r_0$ as, for any $s_{\tau} = (\boldsymbol{t}_{\tau}, \boldsymbol{x}^{\tau-1}, \hat{\boldsymbol{t}}^{\tau-1})$, $r_0(s_{\tau}, a_{\tau}; \boldsymbol{\theta}) = u_0(\boldsymbol{t}_{\tau}, (\boldsymbol{x}^{\tau-1}, g^{\boldsymbol{\theta}}(\hat{\boldsymbol{t}}^{\tau-1}, \boldsymbol{a}_{\tau})))$ and the principal's *payoff function* given policy profile $\boldsymbol{\pi}$ as $U_0(\boldsymbol{\pi}) = \mathbb{E}_{H \sim \nu_{\mu}^{\boldsymbol{\pi}, \boldsymbol{\theta}, T-1}} \left[ \sum_{\tau=0}^{T-1} \gamma^{\tau} r_0(S_{\tau}, A_{\tau}; \boldsymbol{\theta}) \right]$.

Therefore, the principal's expected payoff induced by the agents' play in $\mathcal{Y}^{\boldsymbol{\theta}}$ corresponds exactly to her expected payoff in $\mathcal{D}$.

Let $\boldsymbol{\pi}^{\dagger}$ be the policy profile of $\mathcal{Y}^{\boldsymbol{\theta}}$ that corresponds to truthtelling. We claim that $g^{\boldsymbol{\theta}}$ is a BIC and BIR parameterized (direct) dynamic mechanism that maximizes the principal's payoff over the time horizon iff $\boldsymbol{\theta}$ solves the following optimization problem:

$$\max_{\boldsymbol{\theta} \in \Theta} v(\boldsymbol{\theta}) \doteq \mathbb{E}_{H^{\dagger} \sim \nu_{\mu}^{\boldsymbol{\pi}^{\dagger}, \boldsymbol{\theta}, T-1}} \left[ \sum_{\tau=0}^{T-1} \gamma^{\tau} r_0(S_{\tau}^{\dagger}, A_{\tau}^{\dagger}; \boldsymbol{\theta}) \right] \tag{1}$$

$$\text{s.t.} \max_{\boldsymbol{\pi} \in \mathcal{P}^{\mathrm{PO}}} \psi(\boldsymbol{\theta}, \boldsymbol{\pi}) \doteq \sum_{i \in [n]} \mathbb{E}_{\substack{H \sim \nu_{\mu'}^{(\boldsymbol{\pi}_i, \boldsymbol{\pi}_{-i}^{\dagger}), \boldsymbol{\theta}, T-1} \\ H^{\dagger} \sim \nu_{\mu}^{\boldsymbol{\pi}^{\dagger}, \boldsymbol{\theta}, T-1}}} \left[ \sum_{\tau=0}^{T-1} \gamma^{\tau} \left( r_i(S_{\tau}, A_{\tau}; \boldsymbol{\theta}) - r_i(S_{\tau}^{\dagger}, A_{\tau}^{\dagger}; \boldsymbol{\theta}) \right) \right] = 0 \tag{2}$$

$$h_i(\boldsymbol{\theta}) \doteq \mathbb{E}_{H^{\dagger} \sim \nu_{\mu}^{\boldsymbol{\pi}^{\dagger}, \boldsymbol{\theta}, T-1}} \left[ \sum_{\tau=0}^{T-1} \gamma^{\tau} r_i(S_{\tau}^{\dagger}, A_{\tau}^{\dagger}; \boldsymbol{\theta}) \right] \geq 0 \ \ \forall i \in [n] \tag{3}$$

where $v(\boldsymbol{\theta}) = U_0(\boldsymbol{\pi}^{\dagger}; \boldsymbol{\theta})$ is the principal's payoff under $\boldsymbol{\pi}^{\dagger}$, $\psi(\boldsymbol{\theta}, \boldsymbol{\pi}) = \Psi(\boldsymbol{\pi}^{\dagger}, \boldsymbol{\pi}; \boldsymbol{\theta})$ is the cumulative regret of $\boldsymbol{\pi}$ vs. $\boldsymbol{\pi}^{\dagger}$, and $h_i(\boldsymbol{\theta}) = U_i(\boldsymbol{\pi}^{\dagger}; \boldsymbol{\theta})$ is player $i$'s payoff under $\boldsymbol{\pi}^{\dagger}$. In this way, eq. (1) maximizes the principal's expected payoff over the parameter space; eq. (2) enforces incentive compatibility by requiring the *exploitability* of $\boldsymbol{\pi}^{\dagger}$ in $\mathcal{Y}^{\boldsymbol{\theta}}$ to be zero, meaning no agent can gain by unilaterally deviating from truthful reporting; and eq. (3) guarantees individual rationality (i.e., voluntary participation), since each agent's payoff is non-negative.

The intuitive approach to solving this constrained optimization problem is to apply Lagrangian relaxation. However, a major drawback of this method is that the optimal multiplier can become excessively large, complicating the solution process. To address this, we propose an alternative method that solves the optimization problem without requiring computation of the optimal Lagrangian multiplier. Specifically, we transfer the problem to find the largest $\delta \in \mathbb{R}$ s.t.

$$\min_{\boldsymbol{\theta} \in \Theta} \max_{\boldsymbol{\pi} \in \mathcal{P}^{\mathrm{PO}}} f(\boldsymbol{\theta}, \boldsymbol{\pi}; \delta) \doteq |v(\boldsymbol{\theta}) - \delta| + \alpha \psi(\boldsymbol{\theta}, \boldsymbol{\pi}) + \beta h(\boldsymbol{\theta}) = 0 \tag{4}$$

where $h(\boldsymbol{\theta}) \doteq \sum_{i \in [n]} |h_i(\boldsymbol{\theta})|$, $\alpha, \beta \in \mathbb{R}_+$ scales $\psi$ and $h$ respectively. In this formulation, we search over target principal payoffs $\delta$, and for each $\delta$ we test whether a BIC and BIR dynamic mechanism exists by solving the min-max optimization problem in eq. (4). The minimizer selects parameters to achieve the target payoff, ensure individual rationality, and rationalize $\boldsymbol{\pi}^{\dagger}$ by minimizing its exploitability, while the maximizer chooses per-agent deviations to challenge the rationality of $\boldsymbol{\pi}^{\dagger}$.

**Theorem 3.1.** *Let $\delta^* \in \mathbb{R}$ be the largest real number such that*

$$\min_{\boldsymbol{\theta} \in \Theta} \max_{\boldsymbol{\pi} \in \mathcal{P}^{\mathrm{PO}}} f(\boldsymbol{\theta}, \boldsymbol{\pi}; \delta) = |v(\boldsymbol{\theta}) - \delta^*| + \alpha \psi(\boldsymbol{\theta}, \boldsymbol{\pi}) + \beta h(\boldsymbol{\theta}) = 0,$$

*with $(\boldsymbol{\theta}^*, \boldsymbol{\pi}^*; \delta)$ being the min-max solution, i.e., $f(\boldsymbol{\theta}^*, \boldsymbol{\pi}^*; \delta) = \min_{\boldsymbol{\theta} \in \Theta} \max_{\boldsymbol{\pi} \in \mathcal{P}^{\mathrm{PO}}} f(\boldsymbol{\theta}, \boldsymbol{\pi}; \delta)$, then $g^{\boldsymbol{\theta}^*}$ is the optimal BIC and BIR dynamic mechanism in the mechanism class $\mathcal{G}^{\Theta}$, and $\delta^*$ corresponds to the optimal principal payoff.*

# 4 COMPUTATIONAL RESULTS

In this section, we present an algorithm for the problem introduced above and describe two approaches to addressing the partial observability in the min–max optimization (eq. (4)): one that, under suitable assumptions, guarantees polynomial-time convergence in the discrete-type, compressed-information setting, and another that applies more broadly to general scenarios.

We use a binary-search-like procedure to search for target principal payoff $\delta$. Starting with an interval $[\ell, u]$, the algorithm repeatedly bisects the interval and calls a two-timescale SGDA routine (Algorithm 1, (Daskalakis et al., 2021)) to solve for eq. (4) on the midpoint $c = \frac{l+u}{2}$. If the routine finds a parameter $\boldsymbol{\theta}$ that makes the objective nearly feasible (value $\leq \varepsilon$), we increase the lower bound to $c$; otherwise, we decrease the upper bound. This process continues until the interval is within precision $\varepsilon$, at which point the last feasible $\boldsymbol{\theta}$ is returned.

---

**Algorithm 1** Two-Timescale SGDA

---

**Require:** $\Theta, \mathcal{W}, \eta_{\boldsymbol{\theta}}, \eta_{\boldsymbol{w}}, T, \boldsymbol{\theta}^{(0)}, \boldsymbol{w}^{(0)}, \boldsymbol{w}^{\dagger}$
**Ensure:** $(\boldsymbol{\theta}^{(t)}, \boldsymbol{w}^{(t)})_{t=0}^{T}$

1: Build gradient estimator $\widehat{f}$ associated with $\mathcal{Y}^{\boldsymbol{\theta}}$.
2: **for** $t = 0, \ldots, T-1$ **do**
3:      Sample $\boldsymbol{H} \sim \bigtimes_{i \in [n]} \nu_{\mu}^{(w_i^{(t)}, w_{-i}^{\dagger}), \boldsymbol{\theta}^{(t)T-1}}$,
     $H^{\dagger} \sim \nu_{\mu}^{\boldsymbol{w}^{\dagger}, \boldsymbol{\theta}^{(t)}, T-1}$
4:      $\boldsymbol{\theta}^{(t+1)} \leftarrow \Pi_{\Theta}\left[\boldsymbol{\theta}^{(t)} - \eta_{\boldsymbol{\theta}}^{(t)} \widehat{f_{\boldsymbol{\theta}}}(\boldsymbol{\theta}^{(t)}, \boldsymbol{w}^{(t)}; \boldsymbol{H}, H^{\dagger})\right]$
5:      $\boldsymbol{w}^{(t+1)} \leftarrow \Pi_{\mathcal{P}}\left[\boldsymbol{w}^{(t)} + \eta_{\boldsymbol{w}}^{(t)} \widehat{f_{\boldsymbol{w}}}(\boldsymbol{\theta}^{(t)}, \boldsymbol{w}^{(t)}; \boldsymbol{H}, H^{\dagger})\right]$
6: **end for**
7: **return** $(\boldsymbol{\theta}^{(t)}, \boldsymbol{w}^{(t)})_{t=0}^{T}$

---

Although the binary-search framework reduces the optimization problem to solving a sequence of feasibility checks, the central difficulty lies in solving the min–max objective itself. First, we note that $\boldsymbol{\pi} \in \mathcal{P}^{\text{PO}}$ is a function with a continuous domain. Therefore, as is usual in reinforcement learning, we want to use policy gradient to solve the maximizer's problem. To do so, we restrict the maximizer's space to a policy class $\mathcal{P}^{\mathcal{W}}$ parameterized by $\mathcal{W} \subseteq \mathbb{R}^l$. Redefining $\psi(\boldsymbol{\theta}, \boldsymbol{w}) \doteq \psi(\boldsymbol{\theta}, \boldsymbol{\pi}^{\boldsymbol{w}})$ for all $\boldsymbol{\pi}^{\boldsymbol{w}} \in \mathcal{P}^{\mathcal{W}}$, we aim to solve the problem

$$\min_{\boldsymbol{\theta} \in \Theta} \max_{\boldsymbol{w} \in \mathcal{W}} f(\boldsymbol{\theta}, \boldsymbol{w}; \delta) \doteq |v(\boldsymbol{\theta}) - \delta| + \alpha\psi(\boldsymbol{\theta}, \boldsymbol{w}) + \beta h(\boldsymbol{\theta}). \tag{5}$$

Running Algorithm 1 on $f$ requires an estimate of $\nabla f$ w.r.t. both $\boldsymbol{\theta}$ and $\boldsymbol{w}$. As both gradients involves expectations over histories, we assume that we can simulate trajectories of play from the deviation history distribution $\boldsymbol{H} \doteq (H^1, \ldots, H^n)^T \sim \bigtimes_{i \in [n]} \nu_{\mu}^{(\boldsymbol{\pi}_i^{\boldsymbol{w}}, \pi_{-i}^{\dagger}), \boldsymbol{\theta}, T-1}$ and the truthful reporting history distribution $H^{\dagger} \sim \nu_{\mu}^{\boldsymbol{\pi}^{\dagger}, \boldsymbol{\theta}, T-1}$, and that doing so provides both value and gradient information for the rewards and transition probabilities along simulated trajectories. That is, we rely on a differentiable game simulator (see, for instance, Suh et al. (2022)), a stochastic first-order oracle that returns the gradients of the rewards and transition probabilities, which we query to estimate $\nabla f$.

A central challenge in sampling history trajectories is the *partial observability* of the game: agents never observe the full state, as each agent's type evolves stochastically and remains hidden from both the principal and the other agents. Policies must therefore depend only on private information $\iota_{i,\tau}$, making standard RL methods inapplicable. To address this, we introduce two complementary representations of private information: (i) explicit Bayesian belief, which reduce the POMG to a Markov game in the discrete-type setting (Section 4.1), and (ii) recurrent neural encoders, which embed agent-specific private information through RNNS and scale to continuous types and complex environments where exact inference is infeasible (Section D.1).

## 4.1 BELIEF-BASED REPRESENTATION OF PRIVATE INFORMATION

Our first approach represents beliefs explicitly by maintaining Bayesian distributions over the state space. Although types are hidden, their distributions evolve predictably given the prior, the transition kernel, and past outcomes, so Bayes' rule reduces the POMG to a Markov game (MG). At each step $\tau \in [(T-1)^*]$, each player's belief $b_{i,\tau}$ is a *sufficient statistic* for their private information, and the joint belief forms the belief state $\boldsymbol{b}_\tau$ (formal reduction in Appendix B.1.2). We assume finite type spaces so that beliefs lie in a simplex,[5] and note that public outcomes suffice for belief updates since transitions depend on reports only through induced outcomes. Finally, we focus on *infinite-horizon*

---

[5]Extensions to continuous types require approximations such as particle filtering or function approximation, trading off precision and tractability.

environments, enabling the use of modern RL results (Bhandari & Russo, 2022; Daskalakis et al., 2021) for polynomial-time convergence guarantees.

Before proceeding to the reduction, we must address a challenge that prevents polynomial-time convergence: the dimensionality of the state space in $\mathcal{Y}^{\boldsymbol{\theta}}$. Since each state encodes the entire history of joint outcomes and reports, the state space becomes uncountable in the infinite-horizon setting. To manage this, we assume there exist *compression maps* $\Phi_{\boldsymbol{x}} : \bigcup_{\tau=0}^{T-1} \mathcal{X}^{\tau} \to \mathbb{R}^{d_{\boldsymbol{x}}}$, $\Phi_{\hat{\boldsymbol{t}}} : \bigcup_{\tau=0}^{T-1} \mathcal{T}^{\tau} \to \mathbb{R}^{d_{\hat{\boldsymbol{t}}}}$ that summarize past outcome and report histories into fixed-dimensional vectors, and corresponding *forward compression maps* $\overrightarrow{\Phi_{\boldsymbol{x}}} : (\Phi_{\boldsymbol{x}}(\boldsymbol{x}^{\tau}), \boldsymbol{x}_{\tau+1}) \mapsto \Phi_{\boldsymbol{x}}(\boldsymbol{x}^{\tau+1})$, $\overrightarrow{\Phi_{\hat{\boldsymbol{t}}}} : (\Phi_{\hat{\boldsymbol{t}}}(\hat{\boldsymbol{t}}^{\tau}), \hat{\boldsymbol{t}}_{\tau+1}) \mapsto \Phi_{\hat{\boldsymbol{t}}}(\hat{\boldsymbol{t}}^{\tau+1})$, that propagate these compressed representations forward as new outcomes and reports are realized. Moreover, we assume agent's and principal's immediate reward and agents' type transition kernels depend on past joint outcomes $\boldsymbol{x}^{\tau-1}$ only through $\Phi_{\boldsymbol{x}}(\boldsymbol{x}^{\tau-1})$; every dynamic mechanism $g$ depend on past report profiles $\hat{\boldsymbol{t}}^{\tau-1}$ only through $\Phi_{\hat{\boldsymbol{t}}}(\hat{\boldsymbol{t}}^{\tau-1})$ [6], so we just need to keep track of $\Phi_{\boldsymbol{x}}(\boldsymbol{x}^{\tau-1}), \Phi_{\hat{\boldsymbol{t}}}(\hat{\boldsymbol{t}}^{\tau-1})$ in the state space. For instance, in bandit auctions (see Section 5), each buyer's type transition depends only on their total past allocation count, so we can define $\Phi_{\boldsymbol{x}}(\boldsymbol{x}^{\tau}) = \sum_{k=0}^{\tau} \boldsymbol{x}_k$, which implies only the cumulative sum of past outcomes is retained.

In this way, some information is inevitably lost, but we retain precisely the statistics that are payoff-relevant for the mechanism and the agents, thereby avoiding an explosion in the state space.

Now, for any $g^{\boldsymbol{\theta}} \in \mathcal{G}^{\Theta}$, let $\mathcal{CPOM}^{\boldsymbol{\theta}}$ denote the compressed POMG defined above, which we call the *Compressed Agent POMG*. Its Markov game reduction is given by $\mathcal{M}^{\boldsymbol{\theta}} \doteq (n, T, \mathcal{B}, \mathcal{A}, P', \gamma, \mu', r', \boldsymbol{\theta}, \Theta)$, which we refer to as the *Compressed Agent Belief MG* $\mathcal{M}^{\boldsymbol{\theta}}$ (see Section B.1.2 for details). We denote the principal payoff and player payoffs in this game by $U_0'$ and $(U_i')_{i \in [n]}$ respectively, and we denote the players' action values under policy profile $\boldsymbol{\pi}$ by $q_i^{\boldsymbol{\pi}}$.

For any $i \in [n]$ and any deterministic policy $\boldsymbol{\pi}_i \in \mathcal{P}_i^{\mathrm{PO}}$ in $\mathcal{CPOM}^{\boldsymbol{\theta}}$, there exists an equivalent Markov policy $\boldsymbol{\pi}_i' \in \mathcal{P}_i^{\mathrm{Markov}}$ in the corresponding Belief MG $\mathcal{M}^{\boldsymbol{\theta}}$. Therefore, we translate the maximization problem $\max_{\boldsymbol{\pi} \in \mathcal{P}^{\mathrm{PO}}} \psi(\boldsymbol{\theta}, \boldsymbol{\pi})$ to $\max_{\boldsymbol{\pi} \in \mathcal{P}^{\mathrm{Markov}}} \psi(\boldsymbol{\theta}, \boldsymbol{\pi})$, Without loss of generality[7], we can further restrict to the set of Markov stationary policy profiles $\mathcal{P}^{\mathrm{MS}}$. We overload notation by parameterizing $\mathcal{P}^{\mathrm{MS}}$ directly by $\mathcal{W}$, so each $\boldsymbol{\pi}^{\boldsymbol{w}}$ is a randomized Markov stationary policy profile in $\mathcal{M}^{\boldsymbol{\theta}}$.

Without any additional assumptions, $f(\boldsymbol{\theta}, \boldsymbol{w}; \delta) = |v(\boldsymbol{\theta}) - \delta| + \alpha\psi(\boldsymbol{\theta}, \boldsymbol{w}) + \beta h(\boldsymbol{\theta})$ is in general non-convex-non-concave, and even non-smooth in $\boldsymbol{\theta}$, which makes the associated min–max optimization problem highly challenging. Without additional structural assumptions, first-order methods lack polynomial-time guarantees for finding even $\varepsilon$-stationary solution.

So first, to handle the non-smoothness introduced by absolute values, we approximate the absolute values through a smooth surrogate, namely the $\kappa$-*scaled pseudo-Huber function* $\phi_{\kappa}$ defined as $\phi_{\kappa}(x) = \sqrt{x^2 + \kappa^2} - \kappa$ for all $x \in \mathbb{R}$, with smoothing parameter $\kappa > 0$. Thus, our *smoothed objective function* becomes $f_{\kappa}(\boldsymbol{\theta}, \boldsymbol{w}; \delta) = \phi_{\kappa}(v(\boldsymbol{\theta}) - \delta) + \alpha\psi(\boldsymbol{\theta}, \boldsymbol{w}) + \sum_{i \in [n]} \phi_{\kappa}(h_i(\boldsymbol{\theta}))$.

By imposing additional conditions on policy parametrization, mechanism parameterization, and the original DMD problem (Assumption 1-3, Appendix C), we can ensure that our objective $f$, and hence $f_{\kappa}$, is gradient-dominated in $\boldsymbol{w}$, and thus obtain polynomial-time convergence to an approximate stationary point of *smoothed max-value function* $V_{\kappa}(\boldsymbol{\theta}) \doteq \max_{\boldsymbol{w} \in \mathcal{W}} f_{\kappa}(\boldsymbol{\theta}, \boldsymbol{w}; \delta)$, which corresponds to an first-order locally-optimal BIC+BIR mechanism (Theorem 4.1, (a)). Assumption 1 has three roles: it guarantees the policy parameterization is expressive enough to capture best responses, requires smoothness of the mapping from parameters to actions (and thus of the objective), and imposes structural conditions on $\mathcal{M}$ that yield gradient dominance of the objective. Assumption 2 places analogous constraints on the mechanism parameter space, but only requires smoothness of the

---

[6] For every agent $i$, we redefine $u_i(t_{i,\tau}, \boldsymbol{x}^{\tau})$ as $u_i(t_{i,\tau}, \Phi_{\boldsymbol{x}}(\boldsymbol{x}^{\tau-1}), \boldsymbol{x}_{\tau})$, $F_i(t_{i,\tau+1} \mid t_{i,\tau}, \boldsymbol{x}^{\tau})$ as $F_i(t_{i,\tau+1} \mid t_{i,\tau}, \Phi_{\boldsymbol{x}}(\boldsymbol{x}^{\tau-1}), \boldsymbol{x}^{\tau})$, and for the principal, we redefine $u_0(t_{0,\tau}, \boldsymbol{x}^{\tau})$ as $u_0(\boldsymbol{t}_{\tau}, \Phi_{\boldsymbol{x}}(\boldsymbol{x}^{\tau-1}), \boldsymbol{x}_{\tau})$. Finally, for any dynamic mechanism $g$, we redefine $g(\hat{\boldsymbol{t}}^{\tau})$ as $g(\Phi_{\hat{\boldsymbol{t}}}(\hat{\boldsymbol{t}}^{\tau-1}), \hat{\boldsymbol{t}}^{\tau})$.

[7] Since $\psi$ decomposes into $n$ separate policy optimization problems—one for each player deviating while others remain truthful—each deviation reduces to solving a stationary infinite-horizon discounted belief-MDP. By standard MDP theory (Puterman, 1994), such problems admit optimal policies that are stationary (possibly randomized).

mapping from mechanism parameters to outcomes. Finally, Assumption 3 assumes smooth reward functions and type-transition kernels, further ensuring smoothness of the objective.

Moreover, if the DMD environment is contextual-bandit–like—that is, transitions are independent of outcomes (Assumption 4, Appendix C)—then under additional assumptions on the mechanism parameterization, we obtain polynomial-time convergence to the min–max solution $(\boldsymbol{\theta}^*, \boldsymbol{w}^*)$ of $f$, where $\boldsymbol{\theta}^*$ corresponds to the globally optimal BIC, BIR mechanism (Theorem 4.1(b)).

Finally, we define the *equilibrium distribution mismatch coefficient* $\|\partial \delta_\mu^{\boldsymbol{\pi}^\dagger, \boldsymbol{\theta}} / \partial \mu\|_\infty$ as the Radon-Nikodym derivative of the state occupancy distribution of the truthful-reporting profile $\boldsymbol{\pi}^\dagger$ w.r.t. the initial state distribution $\mu$. This coefficient, which measures the inherent difficulty of reaching states under $\boldsymbol{\pi}^\dagger$, is closely related to other distribution mismatch coefficients introduced in the analysis of policy gradient methods (Agarwal et al., 2020). With this definition in hand, we can finally show polynomial-time convergence of two-timescale stochastic GDA (Algorithm 1).

**Theorem 4.1.** *Suppose Assumption 1-3 hold. For any $\varepsilon \in (0,1)$, if Algorithm 1 is running on $f_\kappa$ with inputs that satisfy $\eta_{\boldsymbol{\theta}}, \eta_{\boldsymbol{w}} \asymp \mathrm{poly}(\varepsilon, \|\partial \delta_\mu^{\boldsymbol{\pi}^*} / \partial \mu\|_\infty, \frac{1}{1-\gamma}, \ell_{\nabla f_\kappa}^{-1}, \ell_{f_\kappa}^{-1})$, then there exists $T \in \mathrm{poly}\left(\varepsilon^{-1}, (1-\gamma)^{-1}, \|\partial \delta_\mu^{\boldsymbol{\pi}^*} / \partial \mu\|_\infty, \ell_{\nabla f_\kappa}, \ell_{f_\kappa}, \mathrm{diam}(\Theta \times \mathcal{W}), \eta_{\boldsymbol{\theta}}^{-1}\right)$ and $k \le T$ s.t.*

*(a)* $\boldsymbol{\theta}_{\mathrm{best}}^{(T)} = \boldsymbol{\theta}^{(k)}$ *is a $(\varepsilon, \varepsilon/2\ell_{f_\kappa})$-stationary point of $V_\kappa$, i.e., there exists $\boldsymbol{\theta}^* \in \Theta$ s.t. $\|\boldsymbol{\theta}_{\mathrm{best}}^{(T)} - \boldsymbol{\theta}^*\| \le \varepsilon/2\ell_{f_\kappa}$ and $\min_{\boldsymbol{h} \in \mathcal{D}V_\kappa(\boldsymbol{\theta}^*)} \|\boldsymbol{h}\| \le \varepsilon$.*

*(b) Moreover, if we further assume that Assumption 4 holds and for all $g^{\boldsymbol{\theta}} \in \mathcal{G}^\Theta$, $\boldsymbol{\theta} \mapsto g^{\boldsymbol{\theta}}(\hat{\boldsymbol{t}}^\tau)$ is affine for all $\tau \in [(T-1)^*]$, $\hat{\boldsymbol{t}}^\tau \in \mathcal{T}^{\tau+1}$, $\boldsymbol{\theta}_{\mathrm{best}}^{(T)}$ satisfies that $\max_{\boldsymbol{w} \in \mathcal{W}} f_\kappa(\boldsymbol{\theta}_{\mathrm{best}}^{(T)}, \boldsymbol{w}) - \min_{\boldsymbol{\theta} \in \Theta} \max_{\boldsymbol{w} \in \mathcal{W}} f_\kappa(\boldsymbol{\theta}, \boldsymbol{w}; \delta) \le \varepsilon$. Furthermore, $\max_{\boldsymbol{w} \in \mathcal{W}} f(\boldsymbol{\theta}_{\mathrm{best}}^{(T)}, \boldsymbol{w}; \delta) - \min_{\boldsymbol{\theta} \in \Theta} \max_{\boldsymbol{w} \in \mathcal{W}} f(\boldsymbol{\theta}, \boldsymbol{w}; \delta) \le \varepsilon + (n+1)\kappa$.*

### 4.2 Neural Embedding of Private Information

While explicit belief tracking provides a principled reduction from POMGs to MGs, it becomes intractable in continuous type spaces: maintaining exact beliefs requires operating in infinite-dimensional spaces, which is computationally prohibitive. To address this limitation, we adopt a recurrent neural architecture for information representation.

It consists of two complementary components: a *public encoder*, which processes commonly observed signals such as the normalized time step and past joint outcomes into a public embedding $c_\tau$, and a collection of *private encoders*, one for each agent, which maintain agent-specific hidden states $(h_{i,\tau})_{i \in [n]}$ by fusing private observations (past types and reports) with the evolving public embedding. For each agent $i$, the pair $(c_\tau, h_{i,\tau})$ serves as a *sufficient statistics* of the information available at time $\tau$, i.e., $\iota_{i,\tau}$, and serves as input to the policy network. This design follows common practice in POMDPs and multi-agent RL (Hausknecht & Stone, 2017; Venkatraman et al., 2017; Foerster et al., 2016), where recurrent architectures are employed to capture and update information over time (see Appendix D.1 for details).

## 5 Experiments: Optimal Bandit Auction

We evaluate our framework in *Bandit Auction* environments, a canonical example of dynamic mechanism design where buyers' types evolve through a bandit process and the principal repeatedly chooses allocations (incurring costs) and payments. The bandit auction admits closed-form optimal allocation rules via virtual index policies, making it a practical and interpretable testbed. We study both discrete and continuous type spaces, using belief-based representations in the former and neural embeddings in the latter. In each case, we consider single-item auctions—benchmarked against the unidimensional results of Pavan et al. (2014)—and multi-item auctions, where no benchmark solutions exist. This design validates our methods in tractable single-item settings and demonstrates scalability to more complex multi-item environments[8].

**Bandit .** The *Bandit Auction* is a setting where a profit-maximizing seller runs a sequence of auctions over $T$ periods, selling $m$ indivisible, non-storable goods to $n$ buyers whose valuations evolve when they win goods. This captures applications such as repeated sponsored search auctions, where

---

[8]Code available at here.

advertisers update valuations based on click profitability. A *bandit auction* can be viewed as a DMD problem defined in Appendix D.2.

**Experiment Description.** We evaluate our framework across four settings (discrete vs. continuous types; single vs. multi-item), using $n = 2$, horizon $T = 5$, and $m = 2$ in multi-item auctions. Valuation spaces and transition functions are specified in Table 2, Appendix D.3. Any mechanism parameter $\boldsymbol{\theta}$ is assessed by **profit loss** $|v(\boldsymbol{\theta}) - \delta|$, **IR loss** $\sum_i |h_i(\boldsymbol{\theta})|$, and **exploitability** $\max_{\boldsymbol{\pi} \in \mathcal{P}^{\text{PO}}} \psi(\boldsymbol{\pi}, \boldsymbol{\theta})$. Our training proceeds in two stages: (i) a hyperparameter search over learning rates and scaling coefficients $(\alpha, \beta)$; and (ii) with the best configuration, running Algorithm 1 for up to 10,000 epochs per profit target $\delta$, retaining mechanisms only when all three losses are below 0.1.

**Network Architecture.** We provide a high-level overview of the mechanism and policy network architectures here, with full details deferred to Appendix D.4.

*Mechanism Network.* Our *mechanism network* takes four inputs—current types, current reports, cumulative allocations, and past reports. Each is encoded by a dedicated MLP, with report histories summarized via attention. The embeddings are concatenated into a shared representation from which allocation and payment heads are derived. Key features (types, reports, allocations) use larger embeddings, while histories provide lower-dimensional context.

*Policy Networks.* Our policy networks use a prototype–residual architecture (Silver et al., 2019; Johannink et al., 2018), with the truthful policy as the prototype (a zero-regret baseline) and a residual network learning profitable deviations.

- Discrete types. Input includes Bayesian beliefs and private information; output is a logit vector over reports. The prototype is the one-hot truthful report, scaled by a learnable parameter, with the residual network adding deviations.
- Continuous types. Input combines recurrent embeddings of private information with the raw (bounded) type. Each coordinate is normalized, mapped into logit space, adjusted by the residual, and projected back via sigmoid and rescaling to produce the final report.

Details are in Appendix D.4.

|  |  |  | $\mathcal{L}_{\text{IR}}$ | Expoit | Max Profit |
|---|---|---|---|---|---|
| Discrete | Single-Item | Benchmark | - | - | 6.5089 |
|  |  | Ours | 0.0975 | 0.0938 | 6.5089 |
|  | Multi-Item | Ours | 0.0215 | 0.0010 | 16.6 |
| Continuous | Single-Item | Benchmark | - | - | 9.7066 |
|  |  | Ours | 0.0685 | 0.0341 | 9.7066 |
|  | Multi-Item | Ours | 0.0425 | 0.0490 | 17.65 |

Table 1: Performance summary.

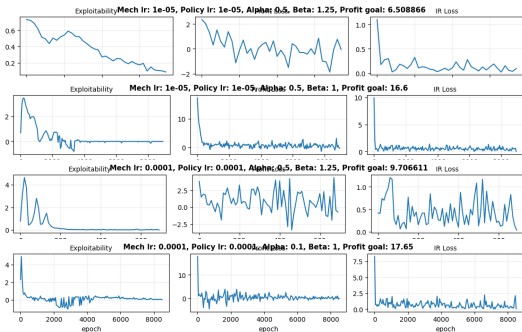

Figure 2: Convergence of exploitability, profit loss, and IR loss (left to right) across four settings: discrete–single, discrete–multi, continuous–single, and continuous–multi (top to bottom)

**Experiment Results.** In the single-item settings, our mechanisms recover the known optimal benchmarks of Pavan et al. (2014), while in the multi-item settings, where analytical solutions continue to elude us, they achieve high profits. Convergence is faster in the single-item case, while the multi-item case requires more training but still reaches low exploitability and IR loss. Table 1 reports the final performance, and Figure 2 illustrates the convergence dynamics across all four settings.

## 6 CONCLUSION

We presented a computational framework for optimal dynamic mechanism design through the inverse-game perspective, casting mechanism design as the search for an optimal inverse truthful equilibrium in a parameterized game space. Our optimization enforces incentive compatibility and individual rationality as equilibrium constraints. Our algorithm provides convergence guarantees for discrete-type settings and handles continuous-type settings with recurrent neural embeddings. Experiments in bandit auctions show that the framework recovers known analytical benchmarks in single-item settings and discovers high-payoff, incentive-compatible mechanisms in multi-item environments without closed-form solutions. Looking forward, we are interested in evaluating the framework beyond profit maximization, incorporating richer objectives (e.g. welfare) to better align with the goals of real-world applications.

