## REFERENCES

Alekh Agarwal, Sham M Kakade, Jason D Lee, and Gaurav Mahajan. Optimality and approximation with policy gradient methods in markov decision processes. In *Conference on Learning Theory*, pp. 64–66. PMLR, 2020. 8

Susan Athey and Ilya Segal. An Efficient Dynamic Mechanism. *Econometrica*, 81(6):2463–2485, 2013. ISSN 1468-0262. doi: 10.3982/ECTA6995. URL https://onlinelibrary.wiley.com/doi/abs/10.3982/ECTA6995. _eprint: https://onlinelibrary.wiley.com/doi/pdf/10.3982/ECTA6995. 14

Moshe Babaioff, Yogeshwer Sharma, and Aleksandrs Slivkins. Characterizing truthful multi-armed bandit mechanisms. In *Proceedings of the 10th ACM conference on Electronic commerce*, pp. 79–88, 2009. 15

Marco Battaglini. Long-Term Contracting with Markovian Consumers. *The American Economic Review*, 95(3):637–658, 2005. ISSN 0002-8282. URL https://www.jstor.org/stable/4132733. Publisher: American Economic Association. 1

Dirk Bergemann and Stephen Morris. Information design: A unified perspective. *Journal of Economic Literature*, 57(1):44–95, 2019. 14

Dirk Bergemann and Juuso Välimäki. The Dynamic Pivot Mechanism. *Econometrica*, 78(2):771–789, 2010. ISSN 1468-0262. doi: 10.3982/ECTA7260. URL https://onlinelibrary.wiley.com/doi/abs/10.3982/ECTA7260. _eprint: https://onlinelibrary.wiley.com/doi/pdf/10.3982/ECTA7260. 14

Dirk Bergemann and Juuso Välimäki. Dynamic Mechanism Design: An Introduction. *Journal of Economic Literature*, 57(2):235–274, 2019. ISSN 0022-0515. URL https://www.jstor.org/stable/26673239. Publisher: American Economic Association. 3, 14

Dimitri P. Bertsekas and Steven E. Shreve. *Stochastic Optimal Control: The Discrete-Time Case | Guide books | ACM Digital Library*. URL https://dl.acm.org/doi/book/10.5555/1512940. 18

Jalaj Bhandari and Daniel Russo. Global Optimality Guarantees For Policy Gradient Methods, June 2022. URL http://arxiv.org/abs/1906.01786. arXiv:1906.01786 [cs, stat]. 7, 19

Mathieu Blondel, Quentin Berthet, Marco Cuturi, Roy Frostig, Stephan Hoyer, Felipe Llinares-López, Fabian Pedregosa, and Jean-Philippe Vert. Efficient and modular implicit differentiation. *arXiv preprint arXiv:2105.15183*, 2021. 23

Simon Board. Durable-Goods Monopoly with Varying Demand. *The Review of Economic Studies*, 75(2):391–413, 2008. ISSN 0034-6527. URL https://www.jstor.org/stable/20185037. Publisher: [Oxford University Press, Review of Economic Studies, Ltd.]. 14

James Bradbury, Roy Frostig, Peter Hawkins, Matthew James Johnson, Chris Leary, Dougal Maclaurin, George Necula, Adam Paszke, Jake VanderPlas, Skye Wanderman-Milne, and Qiao Zhang. JAX: composable transformations of Python+NumPy programs, 2018. URL http://github.com/google/jax. 23

Gianluca Brero, Alon Eden, Matthias Gerstgrasser, David C. Parkes, and Duncan Rheingans-Yoo. Reinforcement Learning of Sequential Price Mechanisms, May 2021. URL http://arxiv.org/abs/2010.01180. arXiv:2010.01180 [cs]. 15

Gianluca Brero, Alon Eden, Darshan Chakrabarti, Matthias Gerstgrasser, Amy Greenwald, Vincent Li, and David C. Parkes. Stackelberg POMDP: A Reinforcement Learning Approach for Economic Design, November 2023. URL http://arxiv.org/abs/2210.03852. arXiv:2210.03852 [cs]. 15

Vincent Conitzer and Tuomas Sandholm. Complexity of mechanism design. In *Proceedings of the Eighteenth conference on Uncertainty in artificial intelligence*, UAI'02, pp. 103–110, San Francisco, CA, USA, August 2002. Morgan Kaufmann Publishers Inc. ISBN 978-1-55860-897-9. 2, 14

Vincent Conitzer and Tuomas Sandholm. Self-interested automated mechanism design and implications for optimal combinatorial auctions. In *Proceedings of the 5th ACM conference on Electronic commerce*, EC '04, pp. 132–141, New York, NY, USA, May 2004. Association for Computing Machinery. ISBN 978-1-58113-771-2. doi: 10.1145/988772.988793. URL https://doi.org/10.1145/988772.988793. 2, 14

Pascal Courty and Li Hao. Sequential Screening. *The Review of Economic Studies*, 67(4):697–717, October 2000. ISSN 0034-6527. doi: 10.1111/1467-937X.00150. URL https://doi.org/10.1111/1467-937X.00150. 1, 14

Michael Curry, Tuomas Sandholm, and John Dickerson. Differentiable economics for randomized affine maximizer auctions. *arXiv preprint arXiv:2202.02872*, 2022. 14

Michael Curry, Vinzenz Thoma, Darshan Chakrabarti, Stephen McAleer, Christian Kroer, Tuomas Sandholm, Niao He, and Sven Seuken. Automated Design of Affine Maximizer Mechanisms in Dynamic Settings, February 2024. URL http://arxiv.org/abs/2402.08129. arXiv:2402.08129 [cs]. 15

Michael J. Curry, Zhou Fan, Yanchen Jiang, Sai Srivatsa Ravindranath, Tonghan Wang, and David C. Parkes. Automated mechanism design: A survey. *ACM SIGecom Exchanges*, 22(2):102–120, 2025. 15

Constantinos Daskalakis, Paul W Goldberg, and Christos H Papadimitriou. The complexity of computing a nash equilibrium. *SIAM Journal on Computing*, 39(1):195–259, 2009. 1

Constantinos Daskalakis, Dylan J. Foster, and Noah Golowich. Independent Policy Gradient Methods for Competitive Reinforcement Learning, January 2021. URL http://arxiv.org/abs/2101.04233. arXiv:2101.04233 [cs]. 6, 7, 19, 20

Damek Davis, Dmitriy Drusvyatskiy, Kellie J MacPhee, and Courtney Paquette. Subgradient methods for sharp weakly convex functions. *Journal of Optimization Theory and Applications*, 179:962–982, 2018. 19

Zhijian Duan, Haoran Sun, Yurong Chen, and Xiaotie Deng. A scalable neural network for dsic affine maximizer auction design. *Advances in Neural Information Processing Systems*, 36:56169–56185, 2023. 14

Paul Duetting, Vahab Mirrokni, Renato Paes Leme, Haifeng Xu, and Song Zuo. Mechanism Design for Large Language Models, October 2023. URL http://arxiv.org/abs/2310.10826. arXiv:2310.10826 [cs, econ]. 14

Paul Dütting, Zhe Feng, Harikrishna Narasimhan, David C. Parkes, and Sai Srivatsa Ravindranath. Optimal Auctions through Deep Learning: Advances in Differentiable Economics. *Journal of the ACM*, pp. 3630749, November 2023. ISSN 0004-5411, 1557-735X. doi: 10.1145/3630749. URL https://dl.acm.org/doi/10.1145/3630749. 2

Péter Eso and Balázs Szentes. Optimal Information Disclosure in Auctions and the Handicap Auction. *The Review of Economic Studies*, 74(3):705–731, 2007. ISSN 0034-6527. URL https://www.jstor.org/stable/4626158. Publisher: [Oxford University Press, Review of Economic Studies, Ltd.]. 1, 14

Alireza Fallah, Michael Jordan, and Annie Ulichney. Fair allocation in dynamic mechanism design. *Advances in Neural Information Processing Systems*, 37:125935–125966, 2024. 14

Zhe Feng, Harikrishna Narasimhan, and David C Parkes. Deep learning for revenue-optimal auctions with budgets. In *Proceedings of the 17th International Conference on Autonomous Agents and Multiagent Systems*, pp. 354–362, 2018. 14

Jakob N. Foerster, Yannis M. Assael, Nando de Freitas, and Shimon Whiteson. Learning to Communicate with Deep Multi-Agent Reinforcement Learning, May 2016. URL http://arxiv.org/abs/1605.06676. arXiv:1605.06676 [cs]. 8

Peter Frazier, David Kempe, Jon Kleinberg, and Robert Kleinberg. Incentivizing exploration. In *Proceedings of the fifteenth ACM conference on Economics and computation*, pp. 5–22, 2014. 15

Denizalp Goktas, Amy Greenwald, Sadie Zhao, Alex Koppel, and Sumitra Ganesh. EFFICIENT INVERSE MULTIAGENT LEARNING. 2024. 1, 4

Noah Golowich, Harikrishna Narasimhan, and David C Parkes. Deep learning for multi-facility location mechanism design. In *Proceedings of the Twenty-seventh International Joint Conference on Artificial Intelligence and the Twenty-third European Conference on Artificial Intelligence*, pp. 261–267, 2018. 14

Amy Greenwald, Jiacui Li, and Eric Sodomka. Approximating Equilibria in Sequential Auctions with Incomplete Information and Multi-Unit Demand. In *Advances in Neural Information Processing Systems*, volume 25. Curran Associates, Inc., 2012. URL https://proceedings.neurips.cc/paper_files/paper/2012/hash/801c14f07f9724229175b8ef8b4585a8-Abstract.html. 14

Charles R. Harris, K. Jarrod Millman, Stéfan J van der Walt, Ralf Gommers, Pauli Virtanen, David Cournapeau, Eric Wieser, Julian Taylor, Sebastian Berg, Nathaniel J. Smith, Robert Kern, Matti Picus, Stephan Hoyer, Marten H. van Kerkwijk, Matthew Brett, Allan Haldane, Jaime Fernandez del Rio, Mark Wiebe, Pearu Peterson, Pierre Gérard-Marchant, Kevin Sheppard, Tyler Reddy, Warren Weckesser, Hameer Abbasi, Christoph Gohlke, and Travis E. Oliphant. Array programming with NumPy. *Nature*, 585:357–362, 2020. doi: 10.1038/s41586-020-2649-2. 23

Matthew Hausknecht and Peter Stone. Deep Recurrent Q-Learning for Partially Observable MDPs, January 2017. URL http://arxiv.org/abs/1507.06527. arXiv:1507.06527 [cs]. 8

Tom Hennigan, Trevor Cai, Tamara Norman, Lena Martens, and Igor Babuschkin. Haiku: Sonnet for JAX, 2020. URL http://github.com/deepmind/dm-haiku. 23

J. D. Hunter. Matplotlib: A 2d graphics environment. *Computing in Science and Engineering*, 9(3): 90–95, 2007. doi: 10.1109/MCSE.2007.55. 23

Tobias Johannink, Shikhar Bahl, Ashvin Nair, Jianlan Luo, Avinash Kumar, Matthias Loskyll, Juan Aparicio Ojea, Eugen Solowjow, and Sergey Levine. Residual Reinforcement Learning for Robot Control, December 2018. URL http://arxiv.org/abs/1812.03201. arXiv:1812.03201 [cs]. 9, 22

Leslie Pack Kaelbling, Michael L. Littman, and Anthony R. Cassandra. Planning and acting in partially observable stochastic domains. *Artificial Intelligence*, 101(1):99–134, May 1998. ISSN 0004-3702. doi: 10.1016/S0004-3702(98)00023-X. URL https://www.sciencedirect.com/science/article/pii/S000437029800023X. 17, 18

Sham Kakade, Ilan Lobel, and Hamid Nazerzadeh. Optimal Dynamic Mechanism Design and the Virtual Pivot Mechanism, March 2013. URL https://papers.ssrn.com/abstract=1782211. 1

Kevin Kuo, Anthony Ostuni, Elizabeth Horishny, Michael J Curry, Samuel Dooley, Ping-yeh Chiang, Tom Goldstein, and John P Dickerson. Proportionnet: Balancing fairness and revenue for auction design with deep learning. *arXiv preprint arXiv:2010.06398*, 2020. 14

Haoming Li, Yumou Liu, Zhenzhe Zheng, Zhilin Zhang, Jian Xu, and Fan Wu. Truthful bandit mechanisms for repeated two-stage ad auctions. In *Proceedings of the 30th ACM SIGKDD Conference on Knowledge Discovery and Data Mining*, pp. 1565–1575, 2024. 15

Tianyi Lin, Chi Jin, and Michael Jordan. On gradient descent ascent for nonconvex-concave minimax problems. In *International Conference on Machine Learning*, pp. 6083–6093. PMLR, 2020. 19

Vahab Mirrokni, Renato Paes Leme, Pingzhong Tang, and Song Zuo. Optimal Dynamic Auctions are Virtual Welfare Maximizers, December 2018. URL http://arxiv.org/abs/1812.02993. arXiv:1812.02993 [cs, econ]. 14

Roger B. Myerson. Optimal Auction Design. *Mathematics of Operations Research*, 6(1):58–73, 1981. ISSN 0364-765X. URL https://www.jstor.org/stable/3689266. Publisher: INFORMS. 1

Mallesh M. Pai and Rakesh Vohra. Optimal Dynamic Auctions and Simple Index Rules. *Mathematics of Operations Research*, 38(4):682–697, 2013. ISSN 0364-765X. URL https://www.jstor.org/stable/24540877. Publisher: INFORMS. 14

Alessandro Pavan. Dynamic Mechanism Design: Robustness and Endogenous Types. In Bo Honoré, Ariel Pakes, Monika Piazzesi, and Larry Samuelson (eds.), *Advances in Economics and Econometrics*, pp. 1–62. Cambridge University Press, 1 edition, November 2017. ISBN 978-1-316-51052-0 978-1-108-22716-2 978-1-108-40000-8. doi: 10.1017/9781108227162.001. URL https://www.cambridge.org/core/product/identifier/CBO9781108227162A010/type/book_part. 14

Alessandro Pavan, Ilya Segal, and Juuso Toikka. Dynamic Mechanism Design: A Myersonian Approach. *Econometrica*, 82(2):601–653, 2014. ISSN 0012-9682. URL https://www.jstor.org/stable/24029270. Publisher: The Econometric Society. 1, 2, 4, 8, 9, 14

Martin L. Puterman. *Markov Decision Processes: Discrete Stochastic Dynamic Programming*. John Wiley & Sons, Inc., USA, 1st edition, 1994. ISBN 978-0-471-61977-2. 7

Sai Srivatsa Ravindranath, Yanchen Jiang, and David C Parkes. Data market design through deep learning. In A. Oh, T. Naumann, A. Globerson, K. Saenko, M. Hardt, and S. Levine (eds.), *Advances in Neural Information Processing Systems*, volume 36, pp. 6662–6689. Curran Associates, Inc., 2023. URL https://proceedings.neurips.cc/paper_files/paper/2023/file/1577ea3eaf8dacb99f64e4496c3ecddf-Paper-Conference.pdf. 15

Maher Said. Sequential auctions with randomly arriving buyers. *Games and Economic Behavior*, 73 (1):236–243, September 2011. ISSN 0899-8256. doi: 10.1016/j.geb.2010.12.010. URL https://www.sciencedirect.com/science/article/pii/S0899825611000054. 14

Tom Silver, Kelsey Allen, Josh Tenenbaum, and Leslie Kaelbling. Residual Policy Learning, January 2019. URL http://arxiv.org/abs/1812.06298. arXiv:1812.06298 [cs]. 9, 22

Takuo Sugaya and Alexander Wolitzky. The Revelation Principle in Multistage Games. *The Review of Economic Studies*, 88(3 (320)):1503–1540, 2021. ISSN 0034-6527. URL https://www.jstor.org/stable/27031806. Publisher: [Oxford University Press, The Review of Economic Studies, Ltd.]. 2, 14

Hyung Ju Suh, Max Simchowitz, Kaiqing Zhang, and Russ Tedrake. Do differentiable simulators give better policy gradients? In *International Conference on Machine Learning*, pp. 20668–20696. PMLR, 2022. 6

Andrea Tacchetti, DJ Strouse, Marta Garnelo, Thore Graepel, and Yoram Bachrach. A neural architecture for designing truthful and efficient auctions. *arXiv preprint arXiv:1907.05181*, 2019. 14

Guido Van Rossum and Fred L Drake Jr. *Python tutorial*. Centrum voor Wiskunde en Informatica Amsterdam, The Netherlands, 1995. 23

Arun Venkatraman, Nicholas Rhinehart, Wen Sun, Lerrel Pinto, Martial Hebert, Byron Boots, Kris M. Kitani, and J. Andrew Bagnell. Predictive-State Decoders: Encoding the Future into Recurrent Networks, September 2017. URL http://arxiv.org/abs/1709.08520. arXiv:1709.08520 [stat]. 8

Gustavo Vulcano, Garrett van Ryzin, and Costis Maglaras. Optimal Dynamic Auctions for Revenue Management. *Management Science*, 48(11):1388–1407, 2002. ISSN 0025-1909. URL https://www.jstor.org/stable/822614. Publisher: INFORMS. 14

Tonghan Wang, Yanchen Jiang, and David C. Parkes. Gemnet: Menu-based, strategy-proof multibidder auctions through deep learning. In *Proceedings of the 25th ACM Conference on Economics and Computation*, EC '24, pp. 1100, New York, NY, USA, 2024. Association for Computing Machinery. ISBN 9798400707049. doi: 10.1145/3670865.3673454. URL https://doi.org/10.1145/3670865.3673454. 15

Tonghan Wang, Yanchen Jiang, and David C Parkes. Bundleflow: Deep menus for combinatorial auctions by diffusion-based optimization. *arXiv preprint arXiv:2502.15283*, 2025. 15

Noah Williams. Persistent private information. *Econometrica*, 79(4):1233–1275, 2011. 14

Jibang Wu, Zixuan Zhang, Zhe Feng, Zhaoran Wang, Zhuoran Yang, Michael I. Jordan, and Haifeng Xu. Sequential Information Design: Markov Persuasion Process and Its Efficient Reinforcement Learning, February 2022. URL http://arxiv.org/abs/2202.10678. arXiv:2202.10678 [cs, econ]. 14

Brian Hu Zhang, Gabriele Farina, Ioannis Anagnostides, Federico Cacciamani, Stephen Marcus McAleer, Andreas Alexander Haupt, Andrea Celli, Nicola Gatti, Vincent Conitzer, and Tuomas Sandholm. Computing Optimal Equilibria and Mechanisms via Learning in Zero-Sum Extensive-Form Games, May 2024. URL http://arxiv.org/abs/2306.05216. arXiv:2306.05216 [cs]. 15

Hanrui Zhang and Vincent Conitzer. Automated Dynamic Mechanism Design, May 2021. URL http://arxiv.org/abs/2105.06008. arXiv:2105.06008 [cs]. 2, 15

## LLM Usage Disclosure

In accordance with ICLR guidelines on the responsible use of large language models (LLMs), we note that LLMs were used exclusively for refining language and improving formatting. They were not used to generate research ideas, mathematical content, theoretical results, or experimental findings. The authors are solely responsible for the accuracy and integrity of all scientific contributions in this work.

## A  Additional related work.

**Foundations and efficiency in DMD.**  For broad surveys of dynamic mechanism design (DMD), see Bergemann & Välimäki (2019) and Pavan (2017). A large efficiency-oriented thread establishes dynamic analogues of Groves/VCG and studies institutional constraints. The dynamic pivot mechanism implements efficient allocations under evolving private information (Bergemann & Välimäki, 2010). Related work formulates allocation and pricing with stochastic arrivals/departures and timing via MDP/online models and sequential auctions (e.g., Greenwald et al., 2012; Said, 2011; Vulcano et al., 2002). Beyond dominant-strategy templates, Athey & Segal (2013) characterize efficient dynamic mechanisms in environments with intertemporal private information, and Pavan et al. (2014) provide a general revelation and envelope approach for dynamic screening. Dynamic revenue maximization in models with time-varying/evolving types (Pai & Vohra, 2013; Mirrokni et al., 2018). Complementary strands analyze dynamic pricing/screening and institutional constraints—including durable-goods monopoly with varying demand over time (Board, 2008), sequential screening (Courty & Hao, 2000), and a revelation principle for multistage games that clarifies the implementability of extensive-form mechanisms (Sugaya & Wolitzky, 2021). Continuous-time dynamic contracting addresses persistent private information and commitment (Williams, 2011). Fairness and design constraints have also been embedded into dynamic settings (Fallah et al., 2024).

**Information design in dynamic environments.**  A complementary literature studies what the designer/principal should *reveal* over time. Canonical treatments of (static) information design include Bergemann & Morris (2019). Dynamic counterparts analyze sequential persuasion and optimal disclosure when actions and beliefs evolve, e.g., Wu et al. (2022); Eso & Szentes (2007). These tools interact with DMD through belief dynamics, experimentation, and commitment to disclosure policies.

**Automated Mechanism Design (AMD)**  Automated mechanism design originated with Conitzer & Sandholm (2002; 2004), who cast mechanism search as a constrained optimization problem and studied computational complexity and constraint generation. More recent approaches utilizes deep learning to scale to larger and more general settings. Duetting et al. (2023) introduced deep neural networks for auction design, improving representational flexibility. This approach, termed *differentiable economics*, has since been extended to budget-constrained bidders (Feng et al., 2018), payment minimization (Tacchetti et al., 2019), multi-facility location (Golowich et al., 2018), fairness-revenue trade-offs (Kuo et al., 2020), affine maximizer auctions (Curry et al., 2022; Duan et al., 2023),

and data markets (Ravindranath et al., 2023). Structural advancements, such as Wang et al. (2024), achieve exact strategy-proofness rather than approximate incentive compatibility. It is also extended to the single-bidder combinatorial auction setting (Wang et al. (2025)). For a comprehensive survey, see Curry et al. (2025).

**Dynamic automated mechanism design (DAMD).** Recent work pushes the AMD viewpoint into dynamic settings. Zhang & Conitzer (2021) introduce Automated Dynamic Mechanism Design with linear-program formulations (and hardness for long horizons). RL-based approaches learn policies for sequential pricing/auctions: Brero et al. (2021) learn sequential posted-price mechanisms; Brero et al. (2023) study Stackelberg-POMDP formulations and sample-based computation of Stackelberg equilibria relevant to dynamic market design. Parallel efforts connect equilibrium computation with learning in auction games, e.g., Zhang et al. (2024). There is also growing interest in automated design for structured mechanism classes such as affine maximizers in dynamic settings (Curry et al., 2024). These works collectively demonstrate that learning-based search over mechanism spaces (with explicit incentive constraints or ex-post compliance) can scale beyond hand-crafted analytic templates.

**Truthful bandits and incentivized exploration.** A complementary line studies *truthful bandit* and incentivized-exploration mechanisms in repeated/dynamic markets, clarifying when exploration must be separated from exploitation and how to maintain (approximate) BIC while learning (Babaioff et al., 2009; Frazier et al., 2014; Li et al., 2024). These results inform the limits and design of learn-as-you-go dynamic platforms.

**Positioning relative to this paper.** Relative to these literatures, our approach searches directly over parameterized *dynamic* mechanisms and enforces IC/IR as equilibrium constraints in the induced dynamic game. This provides a complementary route to dynamic AMD that is not tied to a particular analytic template (e.g., dynamic VCG) and is amenable to multidimensional types and unknown dynamics, while also accommodating belief updates and exploration when relevant.

## B ADDITIONAL PRELIMINARY DEFINITIONS

**Notation** We denote by $[n]$ the set of integers $\{1, \ldots, n\}$ and $[n^*]$ by $\{0\} \cup [n]$. Let $\mathcal{X}$ be any set, and let $(\mathcal{X}, \mathscr{B}_{\mathcal{X}})$ denote the associated measurable space, where $\mathscr{B}_{\mathcal{X}}$ is the Borel $\sigma$-algebra on $\mathcal{X}$. Unless otherwise noted, we will take $\mathscr{B}_{\mathcal{X}}$ as the default $\sigma$-algebra on $\mathcal{X}$. Additionally, we denote the orthogonal projection operator onto a set $\mathcal{X}$ by $\Pi_{\mathcal{X}}(\boldsymbol{x}) \doteq \arg\min_{\boldsymbol{y} \in \mathcal{X}} \|\boldsymbol{x} - \boldsymbol{y}\|_2^2$. We define the subdifferential of a function $f : \mathcal{X} \to \mathbb{R}$ at a point $\boldsymbol{a} \in \mathcal{X}$ by $\mathcal{D}f(\boldsymbol{a}) \doteq \{\boldsymbol{h} \mid f(\boldsymbol{x}) \geq f(\boldsymbol{a}) + \boldsymbol{h}^T(\boldsymbol{x} - \boldsymbol{a})\}$, and we denote the derivative operator of a function $\boldsymbol{g} : \mathcal{X} \to \mathcal{Z}$ by $\partial \boldsymbol{g}$.

**Terminology.** Fix any norm $\|\cdot\|$. Given $\mathcal{A} \subset \mathbb{R}^d$, the function $f : \mathcal{A} \to \mathbb{R}$ is said to be $\ell_f$-*Lipschitz-continuous* for some $\ell_f \in \mathbb{R}_+$ iff $\forall \boldsymbol{x}_1, \boldsymbol{x}_2 \in \mathcal{X}, \|f(\boldsymbol{x}_1) - f(\boldsymbol{x}_2)\| \leq \ell_f \|\boldsymbol{x}_1 - \boldsymbol{x}_2\|$. If the gradient of $f$ is $\ell_{\nabla f}$-Lipschitz-continuous for some $\ell_{\nabla f} \in \mathbb{R}_+$, $f$ is called $\ell_{\nabla f}$-*Lipschitz-smooth*. The function $f$ is said to be a $\ell_f$-contraction (resp. non-expansion) if it is Lipschitz-continuous with coefficient $\ell_f < 1$ (resp. $\ell_f = 1$). For $\mathcal{X} \subseteq \mathbb{R}^d$, we say $f$ is $(c, \mu)$–*gradient dominated* over $\mathcal{X}$ if there exist constants $c > 0$ and $\mu \geq 0$ such that $\min_{x' \in \mathcal{X}} f(x') \geq f(x) + \min_{x' \in \mathcal{X}} \left[ c \langle \nabla f(x), x' - x \rangle + {\mu}/{2} \|x - x'\|_2^2 \right], \forall x \in \mathcal{X}$. The function is said to be gradient dominated with degree one if $\mu = 0$ and gradient dominated with degree two if $\mu > 0$.

**Omitted Definition for Partially Observable Markov Games** Given a policy profile $\boldsymbol{\pi} \in \mathcal{P}$ and an initial state distribution $\mu \in \Delta(\mathcal{S}_0)$, we define the $\tau$*th-step history distribution measure* $\nu_\mu^{\boldsymbol{\pi}, \boldsymbol{\theta}, \tau}$ on $\mathcal{H}_\tau$ as: for all $\underline{\mathcal{H}_\tau} = (\underline{\mathcal{S}_0}, \underline{\mathcal{O}_0}, \underline{\mathcal{A}_0}, \cdots, \underline{\mathcal{S}_\tau}, \underline{\mathcal{O}_\tau}, \underline{\mathcal{A}_\tau}) \in \mathscr{B}_{\mathcal{H}_\tau}$, $\nu_\mu^{\boldsymbol{\pi}, \boldsymbol{\theta}, \tau}(\underline{\mathcal{H}_\tau}) = \int_{\underline{\mathcal{S}_0}} \int_{\underline{\mathcal{O}_0}} \int_{\underline{\mathcal{A}_0}} \cdots \int_{\underline{\mathcal{S}_\tau}} \int_{\underline{\mathcal{O}_\tau}} \boldsymbol{\pi}_\tau(\underline{\mathcal{A}_\tau} \mid \boldsymbol{o}_0, \boldsymbol{a}_0, \cdots, \boldsymbol{a}_{\tau-1}, \boldsymbol{o}_\tau) O_\tau(d\boldsymbol{o}_\tau \mid s_\tau) \times P_{\tau-1}(ds_\tau \mid s_{\tau-1}, \boldsymbol{a}_{\tau-1}; \boldsymbol{\theta}) \times \cdots \boldsymbol{\pi}_0(d\boldsymbol{a}_0 \mid \underline{o_0}) O_0(d\boldsymbol{o}_0 \mid s_0) \mu(ds_0)$.

## B.1 Markov Games and Reduction of POMGs to MGs

### B.1.1 (Parametrized) Markov Games

A *(parametrized) Markov game (MG)* is a tuple $\mathcal{M}^{\boldsymbol{\theta}} \doteq (n, T, \mathcal{S}, \mathcal{A}, P, \gamma, \mu, r, \boldsymbol{\theta}, \Theta)$ parametrized by a vector $\boldsymbol{\theta}$ in the parameter space $\Theta \subseteq \mathbb{R}^d$:

- Horizon $T$: A positive integer or $\infty$.
- State space $\mathcal{S}$: $\mathcal{S}$ is a nonempty Borel space.
- Action space $\{\mathcal{A}_i\}_{i\in[n]}$: Each $\mathcal{A}_i$ is a nonempty Borel space, and we denote the space of joint actions by $\mathcal{A} = \bigtimes_{i\in[n]} \mathcal{A}_i$.
- Parameterized transition kernels $P : \mathcal{S} \times s \times \mathcal{A} \times \Theta \to [0,1]$: $P$ is a Borel-measurable stochastic kernel on $\mathcal{S}$ given $\mathcal{S} \times \mathcal{A} \times \Theta$.
- Discount factor $\gamma$.
- Initial state distribution $\mu \in \Delta(\mathcal{S})$: A probability measure on $\mathcal{S}$.
- Parameterized reward functions $\{r_i : \mathcal{S} \times \mathcal{A} \times \Theta \to \mathbb{R}\}_{i\in[n]}$: Each $r_i$ is a Borel-measurable function from $\mathcal{S} \times \mathcal{A} \times \Theta$ to $\mathbb{R}$.

The game initiates at time $\tau = 0$ in some state $s_0 \sim \mu$ drawn from an initial state distribution $\mu$. At each time period $\tau = 0, 1, \cdots, T-1$, each player $i \in [n]$ plays an *action* $a_{i,\tau} \in \mathcal{A}_i$ and receives a *reward* $r_i(s_\tau, \boldsymbol{a}_\tau; \boldsymbol{\theta})$. The game then transitions to a new state $s_{\tau+1} \in \mathcal{S}_{\tau+1}$ with probability $P(ds_{\tau+1} \mid s_\tau, \boldsymbol{a}_\tau; \boldsymbol{\theta})$.

A *history (of play)* $\boldsymbol{h} \in \mathcal{H}_\tau \doteq (\mathcal{S} \times \mathcal{A})^\tau$ of length $\tau \in \mathbb{N}+1$ is a sequence of state-action tuples $\boldsymbol{h}_\tau = (s_k, \boldsymbol{a}_k)_{k=0}^{\tau-1}$, and we denote the space of histories by $\mathcal{H} = \bigcup_{\tau=1}^T \mathcal{H}_\tau$.

A *policy* for player $i \in [n]$ is a sequence $\boldsymbol{\pi}_i = (\pi_{i,0}, \pi_{i,1}, \cdots, \pi_{i,T-1})$ such that for each $\tau$, $\pi_{i,\tau}$ is a universally measurable stochastic kernel on $\mathcal{A}_i$ given $\mathcal{H}_\tau \times \mathcal{S}$. If, for each $\tau$, $\pi_{i,\tau}$ is parameterized only by $s_\tau$, $\boldsymbol{\pi}_i$ is *Markov*. Moreover, if for all $0 \leq \tau \leq T-1$, $s \in \mathcal{S}$, $\pi_{i,\tau}(s) = \pi_{i,}(s)$ for some $\pi_{i,} : \mathcal{S} \to \Delta(\mathcal{A}_i)$, then $\boldsymbol{\pi}_i$ is *stationary*. If, for each $\tau$ and $(\boldsymbol{h}_\tau, s_\tau) \in \mathcal{H}_\tau \times \mathcal{S}$, $\pi_{i,\tau}(da_{i,\tau} \mid \boldsymbol{h}_\tau, s_\tau)$ assigns mass one to some point in $\mathcal{A}_i$, $\boldsymbol{\pi}_i$ is *deterministic*. In this case, by a slight abuse of notation, $\boldsymbol{\pi}_i$ can be considered as a sequence of universally measurable mappings $\pi_{i,\tau} : \mathcal{H}_\tau \times \mathcal{S} \to \mathcal{A}_i$.

We refer the space of for player $i \in [n]$ as $\mathcal{P}$, the space of all *stationary* policies as $\mathcal{P}_i^{\mathrm{S}}$, the space of all *Markov* policies as $\mathcal{P}_i^{\mathrm{Markov}}$, and the space of all *Markov and stationary* policies as $\mathcal{P}_i^{\mathrm{MS}}$. As usual, $\boldsymbol{\pi} \doteq (\boldsymbol{\pi}_1, \ldots, \boldsymbol{\pi}_n) \in \mathcal{P} \doteq \bigtimes_{i\in[n]} \mathcal{P}_i$ denotes a *policy profile*.

Given a policy profile $\boldsymbol{\pi} \in \mathcal{P}$ and an initial state distribution $\mu \in \Delta(\mathcal{S}_0)$, we define the *$\tau$th-step history distribution measure* $\nu_\mu^{\boldsymbol{\pi},\boldsymbol{\theta},\tau}$ on $\mathcal{H}_\tau$ as for all $\underline{\mathcal{H}_\tau} = (\underline{\mathcal{S}_0}, \underline{\mathcal{A}_0}, \cdots, \underline{\mathcal{S}_{\tau-1}}, \underline{\mathcal{A}_{\tau-1}}) \in \mathscr{B}_{\mathcal{H}_\tau}$

$$\nu_\mu^{\boldsymbol{\pi},\boldsymbol{\theta},\tau}(\underline{\mathcal{H}_\tau}) = \int_{\underline{\mathcal{S}_0}} \int_{\underline{\mathcal{A}_0}} \cdots \int_{\underline{\mathcal{S}_{\tau-1}}} \boldsymbol{\pi}_{\tau-1}(\underline{\mathcal{A}_{\tau-1}} \mid s_0, \boldsymbol{a}_0, \cdots, s_{\tau-1}) \tag{6}$$

$$\times P(ds_\tau \mid s_{\tau-1}, \boldsymbol{a}_{\tau-1}; \boldsymbol{\theta}) \times \boldsymbol{\pi}_{\tau-2}(d\boldsymbol{a}_{\tau-2} \mid s_0, \boldsymbol{a}_0, \cdots, s_{\tau-2}) \times \cdots \tag{7}$$

$$\boldsymbol{\pi}_0(d\boldsymbol{a}_0 \mid s_0)\mu(ds_0) \tag{8}$$

Furthermore, given a policy profile $\boldsymbol{\pi} \in \mathcal{P}$ and a initial state distribution $\mu \in \Delta(\mathcal{S}_0)$, we define the *$\tau$th-step discounted state occupancy measure* $\delta_\mu^{\boldsymbol{\pi},\boldsymbol{\theta},\tau}$ on $\mathcal{S}$ as for all $\underline{\mathcal{S}_\tau} \in \mathscr{B}_{\mathcal{S}}$,

$$\delta_\mu^{\boldsymbol{\pi},\boldsymbol{\theta},\tau}(\underline{\mathcal{S}_\tau}) = \int_{\boldsymbol{h}_{\tau+1} \in \mathcal{H}_{\tau+1}:s_\tau \in \underline{\mathcal{S}_\tau}} \nu_\mu^{\boldsymbol{\pi},\boldsymbol{\theta},\tau+1}(d\boldsymbol{h}_{\tau+1}) \tag{9}$$

and $\delta_\mu^{\boldsymbol{\pi},\boldsymbol{\theta}}(\underline{\mathcal{S}_\tau}) = \sum_{\tau=0}^{T-1} \delta_\mu^{\boldsymbol{\pi},\boldsymbol{\theta},\tau}(\underline{\mathcal{S}_\tau})$.

Given a policy profile $\boldsymbol{\pi} \in \mathcal{P}$, for any player $i \in [n]$, $0 \leq \tau \leq \tau - 1$, define the *action-value* function $q_i^{\boldsymbol{\pi}} : \mathcal{S} \times \mathcal{A} \to \mathbb{R}$ as:

$$q_i^{\boldsymbol{\pi}}(s, \boldsymbol{a}; \boldsymbol{\theta}) = r_i(s, \boldsymbol{a}; \boldsymbol{\theta}) + \mathbb{E}\left[\sum_{\tau=0}^{T-1} \gamma^\tau r_i(s_\tau, \boldsymbol{\pi}(s_\tau); \boldsymbol{\theta}) \middle| s_\tau = s, \boldsymbol{a}_\tau = \boldsymbol{a}\right] \tag{10}$$

Given a policy profile $\boldsymbol{\pi} \in \mathcal{P}$, for any player $i \in [n]$, the *utility function* is defined as

$$U_i(\boldsymbol{\pi}; \boldsymbol{\theta}) \doteq \int_{\mathcal{H}_T} \left[ \sum_{\tau=0}^{T-1} \gamma^\tau r_i(s_\tau, \boldsymbol{a}_\tau; \boldsymbol{\theta}) \right] d\nu_\mu^{\boldsymbol{\pi}, \boldsymbol{\theta}, T} \tag{11}$$

$$= \mathop{\mathbb{E}}_{H \sim \nu_\mu^{\boldsymbol{\pi}, \boldsymbol{\theta}, T}} \left[ \sum_{\tau=0}^{T-1} \gamma^\tau r_i(S_\tau, A_\tau; \boldsymbol{\theta}) \right] \tag{12}$$

For any $\boldsymbol{\pi}, \boldsymbol{\pi}' \in \mathcal{P}$, the *cumulative regret* is $\Psi(\boldsymbol{\pi}, \boldsymbol{\pi}'; \boldsymbol{\theta}) = \sum_{i \in [n]} U_i(\boldsymbol{\pi}'_i, \boldsymbol{\pi}_{-i}; \boldsymbol{\theta}) - U_i(\boldsymbol{\pi}; \boldsymbol{\theta})$. Moreover, given a policy profile $\boldsymbol{\pi} \in \mathcal{P}$, the *exploitability* of $\boldsymbol{\pi}$ is $\varphi(\boldsymbol{\pi}; \boldsymbol{\theta}) \doteq \max_{\boldsymbol{\pi}' \in \mathcal{P}} \Psi(\boldsymbol{\pi}, \boldsymbol{\pi}'; \boldsymbol{\theta})$, which represents the sum of the players' maximal unilateral payoff deviations.

As usual, an $\varepsilon$-*Bayesian Nash equilibrium* ($\varepsilon$-BNE) of a Markov Game $\mathcal{M}^{\boldsymbol{\theta}}$ is a policy profile $\boldsymbol{\pi}^* \in \mathcal{P}$ such that for all $i \in [n]$, $U_i(\boldsymbol{\pi}^*; \boldsymbol{\theta}) \geq \max_{\boldsymbol{\pi}_i \in \mathcal{P}_i} U_i(\boldsymbol{\pi}_i, \boldsymbol{\pi}^*_{-i}; \boldsymbol{\theta}) - \varepsilon$; and a Bayesian Nash equilibrium occurs when $\varepsilon = 0$.

### B.1.2 REDUCTION FROM POMGs TO MGs

Belief state reduction is a fundamental technique for solving partially observable Markov decision processes (POMDPs), where the belief state—a probability distribution over hidden states—encodes all relevant information about the system's history Kaelbling et al. (1998). By operating in the belief space, a POMDP can be reformulated as an equivalent Markov decision process (MDP), enabling the use of MDP solution frameworks. In this section, we extend belief state reduction techniques to POMGs, which introduces additional complexity to belief state dynamics as multiple agents interact and respond to each other's actions and strategies.

We limit our discussion to finite-state POMGs, as extending belief state reductions to continuous state spaces poses significant computational challenges. In continuous-state POMDPs, the belief state resides in an infinite-dimensional space, requiring approximation techniques like particle filtering or function approximation, which often lead to substantial trade-offs between precision and tractability. These complexities significantly amplify the computational burden, making finite-state cases a more tractable yet still illustrative focus.

Let $\mathcal{Y}^{\boldsymbol{\theta}} \doteq (n, T, \mathcal{S}, \mathcal{A}, P, \gamma, \mu, r, \mathcal{O}, O, \boldsymbol{\theta}, \Theta)$ be a partially observable Markov game (POMG). Assume that $|\mathcal{S}| < \infty$ is finite.

For any $0 \leq \tau \leq T$, $i \in [n]$, let $b_{i,\tau} \in \Delta(\mathcal{S})$ be a probability distribution over $\mathcal{S}$. Then, given $o_{i,\tau+1} \in \mathcal{O}_i$, $\boldsymbol{a}_\tau \in \mathcal{A}$, we define the *belief transition function* $\mathrm{BT} : \Delta(\mathcal{S}) \times \mathcal{O}_i \times \mathcal{A} \to \Delta(\mathcal{S})$ by

$$\mathrm{BT}_\tau(b_{i,\tau}, o_{i,\tau+1}, \boldsymbol{a}_\tau)(s_{\tau+1}) \doteq \mathbb{P}(s_{\tau+1} \mid o_{i,\tau+1}, \boldsymbol{a}_\tau, b_{i,\tau}) \tag{13}$$

$$= \frac{\mathbb{P}(o_{i,\tau+1} \mid s_{\tau+1}, \boldsymbol{a}_\tau, b_{i,\tau}) \mathbb{P}(s_{\tau+1} \mid \boldsymbol{a}_\tau, b_{i,\tau})}{\mathbb{P}(o_{i,\tau+1} \mid \boldsymbol{a}_\tau, b_{i,\tau})} \tag{14}$$

$$= \frac{O_{i,\tau+1}(o_{i,\tau+1} \mid s_{\tau+1}) \sum_{s \in \mathcal{S}} P(s_{\tau+1} \mid s, \boldsymbol{a}; \boldsymbol{\theta}) b_{i,\tau}(s)}{\sum_{s' \in \mathcal{S}_{\tau+1}} \left( O_{i,\tau+1}(o_{i,\tau+1} \mid s') \sum_{s \in \mathcal{S}} P_\tau(s' \mid s, \boldsymbol{a}; \boldsymbol{\theta}) b_{i,\tau}(s) \right)} \tag{15}$$

Now, we can define the *Belief Markov Game* corresponds to $\mathcal{Y}^{\boldsymbol{\theta}}$:

- Horizon $T$: Adopted from $\mathcal{Y}^{\boldsymbol{\theta}}$.

- State spaces $\mathcal{B} = \bigtimes_{i \in [n]} \Delta(\mathcal{S})$: $\mathcal{B}$ is the set of joint belief states over the POMDP states. For each $i \in [n]$, $b_i \in \Delta(\mathcal{S})$ represents player $i$'s belief of distribution of current state.

- Action spaces $\{\mathcal{A}_i\}_{i \in [n]}$: Adopted from $\mathcal{Y}^{\boldsymbol{\theta}}$.

- Parameterized transition kernels $P' : \mathcal{B} \times \mathcal{B} \times \mathcal{A} \times \Theta \to [0, 1]$ is defined as

$$P'(\boldsymbol{b}_{\tau+1} \mid \boldsymbol{b}_\tau, \boldsymbol{a}_\tau) \doteq \prod_{i \in [n]} \left( \int_{\mathcal{O}_i} \mathbb{1}_{\{b_{i,\tau+1}\}}(\mathrm{BT}_\tau(b_{i,\tau}, o_{i,\tau+1}, \boldsymbol{a}_\tau)) \mathbb{P}(o_{i,\tau+1} \mid \boldsymbol{a}_\tau, b_{i,\tau}) \right)$$

  where $\mathbb{P}(o_{i,\tau+1} \mid \boldsymbol{a}_\tau, b_{i,\tau}) = \sum_{s' \in \mathcal{S}_{\tau+1}} \left( O_{i,\tau+1}(o_{i,\tau+1} \mid s') \sum_{s \in \mathcal{S}} P_\tau(s_{\tau+1} \mid s, \boldsymbol{a}; \boldsymbol{\theta}) b_{i,\tau}(s) \right)$.

- Discount factor $\gamma$: Adopted from $\mathcal{Y}^{\boldsymbol{\theta}}$.

- Initial state distribution $\mu' \in \Delta(\mathcal{B})$ is the probability measure on $\mathcal{B}$ that assign point mass to $\boldsymbol{b}_0 = (\mu, \cdots, \mu)$.

- Parameterized reward functions $\{r_i' : \mathcal{B} \times \mathcal{A} \times \Theta \to \mathbb{R}\}_{i \in [n]}$ is defined as: for each $i \in [n]$, $r_i'(\boldsymbol{b}_\tau, \boldsymbol{a}; \boldsymbol{\theta}) \doteq \sum_{s \in \mathcal{S}} r_i(s, \boldsymbol{a}; \boldsymbol{\theta}) b_{i,\tau}(s)$.

**Remark 1.** *One underlying assumption of belief Markov game formulation is that at each time step $\tau$, the joint action profile $\boldsymbol{a}_\tau$ is publicly observable for all the players. This assumption is essential for players to update their posterior beliefs about the state distribution based on their private observations, private beliefs, and the joint action profile. However, in the* Agent POMG, *we can replace the public report profile with publicly observable joint outcome, as the transition kernel depends on the report profile only through their induced joint outcome.*

Note that for any $i \in [n]$, and any *deterministic* policy $\boldsymbol{\pi}_i \in \mathcal{P}_i^{\mathrm{PO}}$ in $\mathcal{Y}^{\boldsymbol{\theta}}$, we can define an equivalent policy $\boldsymbol{\pi}_i' \in \mathcal{P}_i$ in the corresponding *Belief Markov Game* as: for each $0 \leq \tau \leq T - 1$, $\pi_{i,\tau}'(a_i \mid \boldsymbol{b}) = \mathbb{P}(\boldsymbol{\pi}_i(\iota_{i,\tau}) = a_i \mid B_{i,\tau} = b_i)$ where $B_{i,\tau} \in \Delta(\mathcal{S})$ is the belief of player $i$ derived from $\iota_{i,\tau}$.

The proof of the reduction and more details can be found in (Bertsekas & Shreve; Kaelbling et al., 1998).

## C    OMITTED ASSUMPTIONS, RESULTS AND PROOFS

**Theorem 3.1.** *Let $\delta^* \in \mathbb{R}$ be the largest real number such that*

$$\min_{\boldsymbol{\theta} \in \Theta} \max_{\boldsymbol{\pi} \in \mathcal{P}^{\mathrm{PO}}} f(\boldsymbol{\theta}, \boldsymbol{\pi}; \delta) = |v(\boldsymbol{\theta}) - \delta^*| + \alpha \psi(\boldsymbol{\theta}, \boldsymbol{\pi}) + \beta h(\boldsymbol{\theta}) = 0,$$

*with $(\boldsymbol{\theta}^*, \boldsymbol{\pi}^*)$ being the min-max solution, i.e., $f(\boldsymbol{\theta}^*, \boldsymbol{\pi}^*; \delta) = \min_{\boldsymbol{\theta} \in \Theta} \max_{\boldsymbol{\pi} \in \mathcal{P}^{\mathrm{PO}}} f(\boldsymbol{\theta}, \boldsymbol{\pi}; \delta)$, then $g^{\boldsymbol{\theta}^*}$ is the optimal BIC and BIR dynamic mechanism in the mechanism class $\mathcal{G}^\Theta$, and $\delta^*$ corresponds to the optimal principal payoff.*

*Proof.* Consider any $\delta^* \in \mathbb{R}$, and assume that

$$\min_{\boldsymbol{\theta} \in \Theta} \max_{\boldsymbol{\pi} \in \mathcal{P}^{\mathrm{PO}}} f(\boldsymbol{\theta}, \boldsymbol{\pi}; \delta) = |v(\boldsymbol{\theta}) - \delta^*| + \alpha \psi(\boldsymbol{\theta}, \boldsymbol{\pi}) + \beta h(\boldsymbol{\theta}) = 0$$

That means, there exists $\boldsymbol{\theta}^* \in \Theta$, $\boldsymbol{\pi}^* \in \mathcal{P}^{\mathrm{PO}}$ s.t. $f(\boldsymbol{\theta}^*, \boldsymbol{\pi}^*) = 0$.

Note that for any $\boldsymbol{\theta} \in \Theta$,

$$|v(\boldsymbol{\theta}) - \delta^*| \geq 0, \quad h(\boldsymbol{\theta}) = \sum_{i \in [n]} |h_i(\boldsymbol{\theta})| \geq 0,$$

and moreover,

$$\alpha \psi(\boldsymbol{\theta}, \boldsymbol{\pi}^*) = \alpha \max_{\boldsymbol{\pi}} \psi(\boldsymbol{\theta}, \boldsymbol{\pi})$$

$$\geq \alpha \psi(\boldsymbol{\theta}, \boldsymbol{\pi}^\dagger) = \alpha \mathop{\mathbb{E}}_{\substack{H \sim \nu_{\mu'}^{(\pi_i^\dagger, \pi_{-i}^\dagger), \boldsymbol{\theta}, T-1} \\ H^\dagger \sim \nu_\mu^{\pi^\dagger, \boldsymbol{\theta}, T-1}}} \left[ \sum_{\tau=0}^{T-1} \gamma^\tau \left( r_i(S_\tau, A_\tau; \boldsymbol{\theta}) - r_i(S_\tau^\dagger, A_\tau^\dagger; \boldsymbol{\theta}) \right) \right] = 0$$

Therefore, we can conclude that $|v(\boldsymbol{\theta}^*) - \delta^*| = 0$, $\psi(\boldsymbol{\theta}^*, \boldsymbol{\pi}^*) = 0$, $h_i(\boldsymbol{\theta}) = 0$ for all $i \in [n]$. In other words, $g^{\boldsymbol{\theta}}$ is a BIC and IR mechanism that achieves principal payoff $\delta^*$. $\square$

**Assumption 1** (Conditions on Policy Parameterization). *Assume that the class of parametrized policy $\mathcal{P}^{\mathcal{W}}$ satisfies: 1. (Convex Parameter Space) $\mathcal{W}$ is non-empty, compact, and convex; 2. (Closure Under Policy Improvement) for any $\boldsymbol{\pi} \in \mathcal{P}^{\mathcal{W}}$, there exists $\boldsymbol{\pi}^+ \in \mathcal{P}^{\mathcal{W}}$ s.t. $q_i^{\boldsymbol{\pi}}(\boldsymbol{b}, \boldsymbol{\pi}_i^+(\boldsymbol{b}), \boldsymbol{\pi}_{-i}(\boldsymbol{b}); \boldsymbol{\theta}) = \max_{\boldsymbol{\pi}_i' \in \mathcal{P}_i^S} q_i^{\boldsymbol{\pi}'}(\boldsymbol{b}, \boldsymbol{\pi}_i'(\boldsymbol{b}), \boldsymbol{\pi}_{-i}(\boldsymbol{b}); \boldsymbol{\theta})$, for all $i \in [n]$ and $\boldsymbol{b} \in \mathcal{B}$, $\boldsymbol{\theta} \in \Theta$; 3. (Gradient-dominated action-value) for any $i \in [n]$, $\boldsymbol{w} \mapsto q_i^{\boldsymbol{\pi}}(\boldsymbol{b}, \boldsymbol{\pi}_i^{\boldsymbol{w}}(\boldsymbol{b}), \boldsymbol{\pi}_{-i}(\boldsymbol{b}); \boldsymbol{\theta})$ is $(c, \mu)$-gradient-dominated over $\mathcal{W}$ for any $\boldsymbol{\pi} \in \mathcal{P}^{\mathcal{W}}$, $\boldsymbol{\theta} \in \Theta$; 4. (Smoothness) for any $i \in [n]$, $\boldsymbol{w} \mapsto \boldsymbol{\pi}_i^{\boldsymbol{w}}(\boldsymbol{b})$ is twice continuously differentiable for all $\boldsymbol{b} \in \mathcal{B}$.*

**Assumption 2** (Conditions on Mechanism Parameterization). *Assume that the class of parametrized direct dynamic mechanism $\mathcal{G}^\Theta$ satisfies: 1. (Convex Parameter Space) $\Theta$ is non-empty, compact, and*

*convex; 2. (Smoothness) for all $g^{\boldsymbol{\theta}} \in \mathcal{G}^{\Theta}$, $\boldsymbol{\theta} \mapsto g^{\boldsymbol{\theta}}(\hat{\boldsymbol{t}}^{\tau})$ is twice continuously differentiable for all $\tau \in [(T-1)^*]$, $\hat{\boldsymbol{t}}^{\tau} \in \mathcal{T}^{\tau+1}$.*

**Assumption 3** (Condition on DMD Environment). *Assume that an* DMD *problem satisfies: 1. (Smoothness) $u_{0,\tau}$, $u_{i,\tau}$ for all $i \in [n]$, $F$ are twice continuously differentiable in $\boldsymbol{x}_{\tau}$.*

**Assumption 4** (Contextual Bandit DMD). *Assume that an* ODMD *problem satisfies that, 1. for any $i \in [n]$, $F_i$ is independent of $(\Phi_{\boldsymbol{x}}(\boldsymbol{x}^{\tau-1}), \boldsymbol{x}_{\tau})$ for any $\tau \in [(T-1)^*]$; 2. for any $i \in \{0\} \cup [n]$, $u_i$ is independent of $\Phi_{\boldsymbol{x}}(\boldsymbol{x}^{\tau-1})$ for any $\tau \in [(T-1)^*]$; 3. (Affine in $\boldsymbol{\theta}$ and $\boldsymbol{w}$) for any $\tau \in [(T-1)^*]$, $\boldsymbol{x}_{\tau} \mapsto u_{0,\tau}(\boldsymbol{t}_{\tau}, \Phi_{\boldsymbol{x}}(\boldsymbol{x}^{\tau-1}), \boldsymbol{x}_{\tau})$ is affine for all $\boldsymbol{t}_{\tau} \in \mathcal{T}$, $\boldsymbol{x}^{T-1} \in \mathcal{X}^{\tau}$; $\boldsymbol{x}_{\tau} \mapsto u_{i,\tau}(t_{i,\tau}, \Phi_{\boldsymbol{x}}(\boldsymbol{x}^{\tau-1}), \boldsymbol{x}_{\tau})$ is affine for all $i \in [n]$, $t_{i,\tau} \in \mathcal{T}_i$, $\boldsymbol{x}^{T-1} \in \mathcal{X}^{\tau}$.*

**Theorem 4.1.** *Suppose Assumption 1-3 hold. For any $\varepsilon \in (0,1)$, if Algorithm 1 is running on $f_{\kappa}$ with inputs that satisfy $\eta_{\boldsymbol{\theta}}, \eta_{\boldsymbol{w}} \asymp \operatorname{poly}(\varepsilon, \|\partial \delta_{\mu}^{\boldsymbol{\pi}^*}/\partial \mu\|_{\infty}, \frac{1}{1-\gamma}, \ell_{\nabla f_{\kappa}}^{-1}, \ell_{f_{\kappa}}^{-1})$, then there exists $T \in \operatorname{poly}\left(\varepsilon^{-1}, (1-\gamma)^{-1}, \|\partial \delta_{\mu}^{\boldsymbol{\pi}^*}/\partial \mu\|_{\infty}, \ell_{\nabla f_{\kappa}}, \ell_{f_{\kappa}}, \operatorname{diam}(\Theta \times \mathcal{W}), \eta_{\boldsymbol{\theta}}^{-1}\right)$ and $k \le T$ s.t.*

*(a) $\boldsymbol{\theta}_{\mathrm{best}}^{(T)} = \boldsymbol{\theta}^{(k)}$ is a $(\varepsilon, \varepsilon/2\ell_{f_{\kappa}})$-stationary point of $V_{\kappa}$, i.e., there exists $\boldsymbol{\theta}^* \in \Theta$ s.t. $\|\boldsymbol{\theta}_{\mathrm{best}}^{(T)} - \boldsymbol{\theta}^*\| \le \varepsilon/2\ell_{f_{\kappa}}$ and $\min_{\boldsymbol{h} \in \mathcal{D}V_{\kappa}(\boldsymbol{\theta}^*)}\|\boldsymbol{h}\| \le \varepsilon$.*

*(b) Moreover, if we further assume that Assumption 4 holds and for all $g^{\boldsymbol{\theta}} \in \mathcal{G}^{\Theta}$, $\boldsymbol{\theta} \mapsto g^{\boldsymbol{\theta}}(\hat{\boldsymbol{t}}^{\tau})$ is affine for all $\tau \in [(T-1)^*]$, $\hat{\boldsymbol{t}}^{\tau} \in \mathcal{T}^{\tau+1}$, $\boldsymbol{\theta}_{\mathrm{best}}^{(T)}$ satisfies that $\max_{\boldsymbol{w} \in \mathcal{W}} f_{\kappa}(\boldsymbol{\theta}_{\mathrm{best}}^{(T)}, \boldsymbol{w}) - \min_{\boldsymbol{\theta} \in \Theta} \max_{\boldsymbol{w} \in \mathcal{W}} f_{\kappa}(\boldsymbol{\theta}, \boldsymbol{w}; \delta) \le \varepsilon$. Furthermore, $\max_{\boldsymbol{w} \in \mathcal{W}} f(\boldsymbol{\theta}_{\mathrm{best}}^{(T)}, \boldsymbol{w}; \delta) - \min_{\boldsymbol{\theta} \in \Theta} \max_{\boldsymbol{w} \in \mathcal{W}} f(\boldsymbol{\theta}, \boldsymbol{w}; \delta) \le \varepsilon + (n+1)\kappa$.*

*Proof.* Define $V_{\epsilon}(\boldsymbol{\theta}) = \max_{\boldsymbol{w} \in \mathcal{W}} f_{\epsilon}(\boldsymbol{\theta}, \boldsymbol{w})$ for any $\boldsymbol{\theta} \in \Theta$. First, note that even though $f_{\epsilon}$ is smooth in both $\boldsymbol{\theta}$ and $\boldsymbol{w}$ under our assumption, $V_{\epsilon}$ is not guaranteed to be smooth. Therefore, we consider the Moreau envelope of the it, i.e.,

$$\tilde{V}_{\epsilon}(\boldsymbol{\theta}) \doteq \min_{\boldsymbol{\theta}' \in \Theta}\left\{V_{\epsilon}(\boldsymbol{\theta}') + \ell_{\nabla f_{\epsilon}}\|\boldsymbol{\theta} - \boldsymbol{\theta}'\|^2\right\}$$

as common in the optimization literature (see, for instance, Davis et al. (2018)).

We invoke Theorem 2 of (Daskalakis et al., 2021). Although their result is stated for gradient-dominated-gradient-dominated functions, their proof applies in the more general case of non-convex-gradient-dominated functions.

First, condition 4 of Assumption 1, condition 2 of Assumption 2, and condition 2 of Assumption 3 together guarantees that the $f$ is Lipschitz-smooth w.r.t. $(\boldsymbol{\theta}, \boldsymbol{w})$.

Moreover, these conditions also implies that for any $i \in [n]$, $\boldsymbol{w} \mapsto q_i^{\boldsymbol{\pi}}(\boldsymbol{b}, \boldsymbol{\pi}_i^{\boldsymbol{w}}(\boldsymbol{b}), \boldsymbol{\pi}_{-i}(\boldsymbol{b}); \boldsymbol{\theta})$ is continuously differentiable over $\mathcal{W}$ for any $\boldsymbol{\pi} \in \mathcal{P}^{\mathcal{W}}$, $\boldsymbol{\theta} \in \Theta$. Combine this with condition 2, 3 of Assumption 1, we have that $f$ is $\left(\frac{\|\partial \mu_{\mu, \boldsymbol{\pi}^*}/\partial \mu\|_{\infty}}{1-\gamma} \cdot c, \frac{\|\partial \mu_{\mu, \boldsymbol{\pi}^*}/\partial \mu\|_{\infty}}{1-\gamma} \cdot \mu\right)$-gradient-dominated in $\boldsymbol{w}$, for all $\boldsymbol{\theta} \in \Theta$, by Theorems 2 and 4 of (Bhandari & Russo, 2022).

Finally, under Assumption 1-3, since the mechanism, policy, the reward function, and the transition probability function are all Lipschitz-continuous, the gradient estimator $\hat{f}$ is also Lipschitz-continuous, since $\mathcal{S}$ and $\mathcal{A}$ are compact. Their variance must therefore be bounded, i.e., there exists $\varsigma_{\boldsymbol{\theta}}, \varsigma_{\boldsymbol{w}} \in \mathbb{R}$ s.t. $\mathbb{E}_{\boldsymbol{h}, \boldsymbol{h}'}[\widehat{f_{\boldsymbol{\theta}}}(\boldsymbol{\theta}, \boldsymbol{w}; \boldsymbol{h}, \boldsymbol{h}') - \nabla_{\boldsymbol{\theta}} f(\boldsymbol{\theta}, \boldsymbol{w}; \boldsymbol{h}, \boldsymbol{h}')] \le \varsigma_{\boldsymbol{\theta}}$ and $\mathbb{E}_{\boldsymbol{h}, \boldsymbol{h}'}[\widehat{f_{\boldsymbol{w}}}(\boldsymbol{\theta}, \boldsymbol{w}; \boldsymbol{h}, \boldsymbol{h}') - \nabla_{\boldsymbol{w}} f(\boldsymbol{\theta}, \boldsymbol{w}; \boldsymbol{h}, \boldsymbol{h}')] \le \varsigma_{\boldsymbol{w}}$.

Hence, under our assumptions, the assumptions of Theorem 2 of (Daskalakis et al., 2021) are satisfied. Therefore, $1/T+1 \sum_{\tau=0}^{T}\|\nabla \tilde{V}_{\epsilon}(\boldsymbol{\theta}^{(\tau)})\| \le \varepsilon$. Taking a minimum across all $\tau \in [T]$, we conclude $\left\|\nabla \tilde{V}_{\epsilon}(\boldsymbol{\theta}_{\mathrm{best}}^{(T)})\right\| \le \varepsilon$.

Then, by the Lemma 3.7 of Lin et al. (2020), there exists some $\boldsymbol{\theta}^* \in \Theta$ such that $\|\boldsymbol{\theta}_{\mathrm{best}}^{(T)} - \boldsymbol{\theta}^*\| \le \frac{\varepsilon}{2\ell_{f_{\epsilon}}}$ and $\boldsymbol{\theta}^* \in \Theta_{\varepsilon} \doteq \{\boldsymbol{\theta} \in \Theta \mid \exists \alpha \in \mathcal{D}V_{\epsilon}(\boldsymbol{\theta}), \|\alpha\| \le \varepsilon\}$. That is, $\boldsymbol{\theta}_{\mathrm{best}}^{(T)}$ is a $(\varepsilon, \frac{\varepsilon}{2\ell_{f_{\epsilon}}})$-stationary point of $V_{\epsilon}$.

Now, we further assume that Assumption 4 holds and for all $g^{\boldsymbol{\theta}} \in \mathcal{G}^{\Theta}$, $\boldsymbol{\theta} \mapsto g^{\boldsymbol{\theta}}(\hat{\boldsymbol{t}}^{\tau})$ is affine for all $0 \le \tau \le T-1$, $\hat{\boldsymbol{t}}^{\tau} \in \mathcal{T}^{\tau+1}$, condition 3 of Assumption 4, we know that $f_{\epsilon}$ is affine in $\boldsymbol{\theta}$ as it is sum of multiple affine functions. That is, $f_{\epsilon}$ is $\left(\frac{\|\partial \mu_{\mu, \boldsymbol{\pi}^*}/\partial \mu\|_{\infty}}{1-\gamma}, 0\right)$-gradient-dominated in $\boldsymbol{w}$. Then,

by Theorem 2 of (Daskalakis et al., 2021) again, $V_\epsilon(\boldsymbol{\theta}_{\text{best}}^{(T)}) - \min_{\boldsymbol{\theta}\in\Theta} V_\epsilon(\boldsymbol{\theta}) \leq \varepsilon$. Moreover, note that $0 \leq f(\boldsymbol{\theta}, \boldsymbol{w}) - f_\epsilon(\boldsymbol{\theta}, \boldsymbol{w}) \leq (n+1)\epsilon$ for any $(\boldsymbol{\theta}, \boldsymbol{w}) \in \Theta \times \mathcal{W}$, so $V_\epsilon(\boldsymbol{\theta}_{\text{best}}^{(T)}) \leq V(\boldsymbol{\theta}_{\text{best}}^{(T)}) \leq V_\epsilon(\boldsymbol{\theta}_{\text{best}}^{(T)}) + (n+1)\epsilon$. Thus,

$$V(\boldsymbol{\theta}_{\text{best}}^{(T)}) - \min_{\boldsymbol{\theta}\in\Theta} V(\boldsymbol{\theta}) \leq V_\epsilon(\boldsymbol{\theta}_{\text{best}}^{(T)}) + (n+1)\epsilon - \min_{\boldsymbol{\theta}\in\Theta} V_\epsilon(\boldsymbol{\theta})$$

$$= \varepsilon + (n+1)\epsilon$$

$\square$

## D EXPERIMENT DETAILS

### D.1 DETAILS ON NEURAL EMBEDDING OF PRIVATE INFORMATION

**Encoder Architecture.** Our architecture consists of two class of encoders:

- **Public encoder.** At each time step $\tau$, the public encoder processes information that is commonly observed by all agents, including the normalized time step and the previous joint outcome. Formally, the public observation is $o_{\text{pub},\tau} = [\tau/(T-1), \text{vec}(\boldsymbol{x}_{\tau-1})]$. The public hidden state is then updated via $c_\tau = \text{RNN}_{\text{pub}}(f_{\text{pub}}(o_{\text{pub},\tau}), c_{\tau-1})$ where $f_{\text{pub}}$ is a linear projection and $c_\tau \in \mathbb{R}^{d_{\text{pub}}}$ encodes the shared information available to all agents.
- **Private encoder.** Each agent $i$ additionally maintains a private encoder that processes agent-specific observations: the normalized time step, the agent's current type, and her last report. Formally, $o_{i,\tau} = [\tau/(T-1), t_{i,\tau}, \hat{t}_{i,\tau-1}]$ The private hidden state is updated by fusing the agent's private observation with the current public embedding: $z_{i,\tau} = \sigma\left(W_{\text{priv}}(f_{\text{priv}}(o_{i,\tau}) \oplus f_{\text{pub}\to\text{priv}}(c_\tau))\right)$ and $h_{i,\tau} = \text{RNN}_{\text{priv}}(z_{i,\tau}, h_{i,\tau-1})$ where $f_{\text{priv}}(o_{i,\tau})$ is the private observation embedding, $f_{\text{pub}\to\text{priv}}(c_\tau)$ is the projected public embedding, $W_{\text{priv}}$ is an additional linear layer after concatenation, $\sigma$ is a ReLU activation, and $h_{i,\tau}$ is the updated hidden state.

**Intuition and Use.** Together, the embeddings $(c_\tau, h_{i,\tau})$ serve as an implicit belief state for agent $i$ at time step $\tau$: $c_\tau$ captures publicly observable dynamics, while $h_{i,\tau}$ captures each agent's evolving private information. These embeddings replace explicit beliefs as the input to our policy networks, enabling end-to-end training while preserving belief-like structure.

### D.2 DETAILS ON BANDIT AUCTION

A *bandit auction* problem can be viewed as a DMD problem where at time step $\tau$:
- Buyer $i$'s type is her valuations for goods, i.e., $t_{i,\tau} = \boldsymbol{v}_{i,\tau} = (v_{i0,\tau}, \cdots, v_{im,\tau}) \in \mathbb{R}^m$, where $v_{ij,\tau}$ is buyer $i$'s valuation for good $j \in [m]$. At time step $\tau = 0$, each buyer's valuation is drawn independently from a distribution $\omega_i$.
- The immediate reward of buyer $i$ given her type and outcome is $u_{i,\tau}(t_{i,\tau}, x_{i,\tau}) = \sum_{j\in[m]} v_{ij,\tau} x_{ij,\tau} - p_{i,\tau}$. The immediate reward of the seller is $u_{0,\tau}(\boldsymbol{t}_\tau, \boldsymbol{x}_\tau) = \sum_{i\in[n]} p_i - \sum_{i\in[n],j\in[m]} c_{ij}^\tau x_{ij,\tau}$, where $c_{ij}^\tau \in \mathbb{R}$ is the cost of allocating good $j$ to buyer $i$ at time step $\tau$.
- Buyer $i$ receives outcome $x_{i,\tau} = (\boldsymbol{x}_{i,\tau} = (x_{i0,\tau}, \cdots, x_{im,\tau}), p_{i,\tau})$. where each $x_{ij,\tau} \in [0,1]$ represent the probability of allocating good $j$ to buyer $i$, and $p_{i,\tau} \in \mathbb{R}$ is the monetary transfer buyer $i$ paid to the seller.
- The buyer $i$'s valuation for good $j$ then evolves according to $P_{ij}(v_{ij,\tau+1} \mid v_{ij,\tau}, x_{ij,\tau}) = R_{ij}(v_{ij,\tau+1} - v_{ij,\tau} \mid \sum_{k=0}^\tau x_{ij,k}) \cdot x_{ij,\tau} + \mathbb{1}_{\{v_{ij,\tau+1}\}}(v_{ij,\tau}) \cdot (1 - x_{ij,\tau})$. Intuitively, if buyer $i$ is assigned with good $j$, her valuation for good $j$ evolves according to $R_{ij}$; otherwise, her valuation for good $j$ remain unchanged.

### D.3 DETAILS ON EXPERIMENT DESCRIPTION

### D.4 DETAILS ON MECHANISM AND POLICY NETWORKS

**Mechanism Network** The mechanism network processes four streams of information, reflecting their relative importance: (i) the current type profile (highest importance), (ii) the current report (high

|  | **Discrete Types** | **Continuous Types** |
|---|---|---|
| Valuation Space | $\mathcal{V}_{ij} = \{1.0, 2.0, 3.0\},\ \forall i, j$ | $\mathcal{V}_{ij} = [0, 5],\ \forall i, j$ |
| Initial type distribution | $v_{ij,0} \sim \mathrm{Unif}(\{1.0, 2.0, 3.0\})$ | $v_{ij,0} \sim \mathcal{N}(2.5,\ 0.1^2)$ |
| Transition function | $P_{ij}(v_{ij,\tau} \mid v_{ij,\tau-1}, x_{ij,\tau})$ $= \frac{\exp(-\alpha\|v_{ij,\tau}-v_{ij,\tau-1}\|)}{\sum_{v \in \mathcal{V}_{ij}} \exp(-\alpha\|v-v_{ij,\tau-1}\|)}$ where $\alpha = \frac{1}{2}\sum_{k=0}^{\tau-1} x_{ij,k}$. | 1. $s_{ij} = \log\left(\frac{v^*_{ij,\tau}}{1-v^*_{ij,\tau}}\right)$ where $v^*_{ij,\tau} = \frac{v_{ij,\tau}}{5}$.
2. $s_{ij,\tau+1} \sim \mathcal{N}(s_{ij,\tau}, \sigma_{ij,\tau})$
where $\sigma_{ij,\tau} = (1-x_{ij,\tau})\sigma_{no} + x_{ij,\tau}\frac{1}{1+\alpha}$,
$\sigma_{no} = 0.01, \alpha = \frac{1}{2}\sum_{k=0}^{\tau-1} x_{ij,k}$.
3. $v_{ij,\tau+1} = \mathrm{Sigmoid}(s_{ij,\tau+1}) \cdot 5$ |

Table 2: Summary of valuation space, initial distributions and transition functions in discrete vs. continuous settings.

importance), (iii) the cumulative allocation (high importance), and (iv) the sequence of past reports (lower importance). Each stream is first encoded by a dedicated multilayer perceptron (MLP), and the report-history sequence is aggregated via a simple attention mechanism. The resulting features are concatenated and passed through a shared MLP, which branches into allocation and payment heads.

Formally, we define the encodings

$$h^{\text{type}} = f_{\text{type}}(\boldsymbol{t}_\tau), \quad h^{\text{report}} = f_{\text{report}}(\hat{\boldsymbol{t}}_\tau), \quad h^{\text{alloc}} = f_{\text{alloc}}\Big(\sum_{k=0}^{\tau-1} \boldsymbol{x}_k\Big), \quad h^{\text{report\_hist}} = \mathrm{Attn}\Big(\{f_{\text{hist}}(\hat{\boldsymbol{t}}_k)\}_{k=0}^{\tau-1}\Big),$$

where each $f.$ is a small MLP and $\mathrm{Attn}$ denotes attention-based pooling across the history sequence.

The concatenated feature vector is

$$h_\tau = f_{\text{shared}}([\,h^{\text{type}},\,h^{\text{report}},\,h^{\text{alloc}},\,h^{\text{report\_hist}}\,]),$$

where $f_{\text{shared}}$ is a two-layer MLP. From this shared representation, two task-specific heads are applied:

$$a_\tau = \mathrm{Softmax}(W_{\text{alloc}} h_\tau), \qquad\qquad p_\tau = W_{\text{pay}} f_{\text{pay}}(h_\tau),$$

where $a_\tau \in \mathbb{R}^{(n+1) \times m}$ are the allocation probabilities for each good (including the null buyer which denotes no allocation), and $p_\tau \in \mathbb{R}^n$ are the payments for each buyer.

This design ensures that high-importance features (current type, current report, allocation history) are emphasized with larger embedding dimensions, while the report history contributes lower-dimensional context through attention-based summarization.

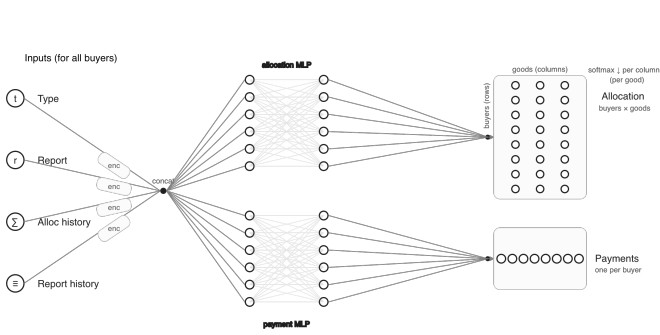

Figure 3: Mechanism network. Inputs are encoded and concatenated, then pass through a shared trunk that branches into an allocation head (buyers+null $\times$ goods; column-wise softmax per good) and a payment head (one scalar per buyer).

**Policy Networks.** We design two policy networks, one for discrete type spaces and one for continuous type spaces, both following a *prototype + residual* structure. This is motivated by residual policy learning in reinforcement learning (Silver et al., 2019; Johannink et al., 2018), where a residual network refines a strong baseline controller. In our setting, the truthful policy is a natural baseline: it achieves zero cumulative regret, so the network should start from truthful reporting and learn deviations only when profitable. This design stabilizes training and anchors optimization around an interpretable solution.

DISCRETE TYPES. For buyer $i$ with type $t_i \in \Theta_i$, let $\mathbf{e}(t_i) \in \{0,1\}^{|\Theta_i|}$ denote the one-hot truthful report. The policy logits are given by

$$\ell_i = k \cdot \mathbf{e}(t_i) + r_i(b),$$

where $k \geq 0$ is a learnable *prototype strength* parameter (initialized large) and $r_i(b) \in \mathbb{R}^{|\Theta_i|}$ is the residual network, which depends on the belief state $b$. The residual is parameterized by

$$r_i(b) = W_{\text{res}}\, \sigma(W_{\text{fuse}}[\, h(b), r_i(b), \mathbf{e}(t_i), \mu_i(b)\,]),$$

where $h(b)$ is a shared embedding of the common information in belief state (time, allocation history), $r_i(b)$ is buyer $i$'s report history, $\mu_i(b)$ is buyer $i$'s belief slice, and $\sigma$ is a ReLU nonlinearity. With zero initialization of $W_{\text{res}}$, the initial policy reduces to truthful reporting.

**Policy — Discrete (single buyer)**

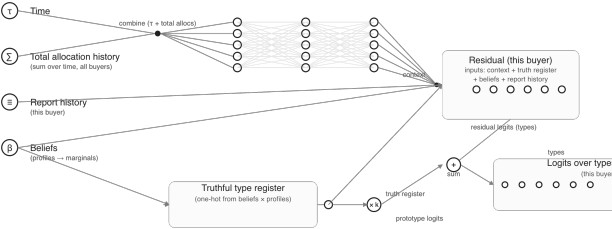

Figure 4: Discrete policy (single buyer). Public context (time + total allocation history) drives a residual head; the truthful type register (from beliefs×profiles) gives prototype logits scaled by $k$. Prototype and residual logits are summed to produce final logits over types for the buyer.

CONTINUOUS TYPES. For bounded continuous types $t_i \in [L_i, H_i]^G$, we normalize each coordinate,

$$p_{ij} = \frac{t_{ij} - L_{ij}}{H_{ij} - L_{ij}}, \quad s_{ij}^0 = \text{logit}(p_{ij}),$$

and add a residual in logit space,

$$s_{ij} = s_{ij}^0 + \Delta_{ij}(h_i, c),$$

where $\Delta_{ij}$ is computed from private and public embeddings $(h_i, c)$. The final report is obtained by mapping back to the original type space,

$$\hat{t}_{ij} = L_{ij} + (H_{ij} - L_{ij}) \cdot \sigma(s_{ij}).$$

As in the discrete case, zero initialization ensures the initial policy is truthful, while $\Delta_{ij}$ learns profitable deviations during training.

**Summary.** Both policies share the same structure: a *truthful prototype* anchored in logit space, plus a *residual network* that learns deviations. This mirrors residual policy learning in control, but is specialized to mechanism design by exploiting the fact that truthful reporting provides a principled baseline with zero regret.

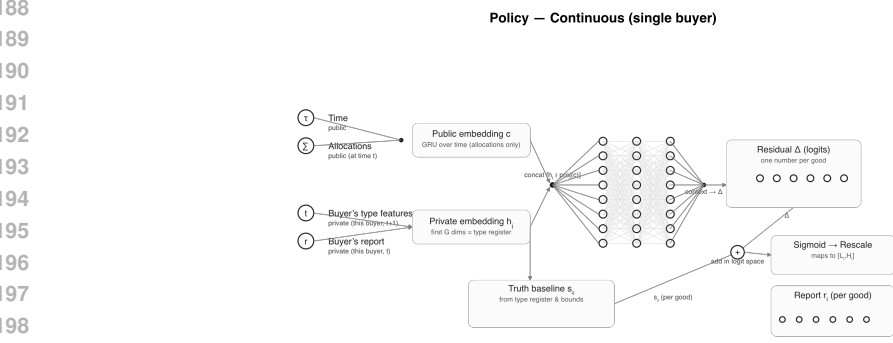

Figure 5: Continuous policy (single buyer). A public embedding summarizes time and allocations; a private embedding uses the buyer's time, next-type features, and current report (with a type register). After concat, a head outputs per-good residual logits $\Delta$, which are added to a baseline $s_0$ from the type register; the sum is passed through sigmoid and rescaled to the bounds to yield $r_i$.

## D.5 DETAILS ON IMPLEMENTATION

We ran four sets of experiments: discrete type–single agent, discrete type–multiple agent, continuous type–single agent, and continuous type–multiple agent.

For each set, we ran Algorithm 1 on the objective function for 1,000 episodes for every combination of learning rate candidates $\eta_\theta, \eta_w$ and scaling parameters $\alpha, \beta$, performing a grid search and measuring performance in terms of exploitability, profit loss, and IR loss. Based on these results, we selected the best hyperparameter combination to use in the final experiments.

In the final stage, we carried out a binary search over the profit target $\delta$. For each value of $\delta$, we ran Algorithm 1 for 10,000 episodes with the selected hyperparameters (shown in Figure 2), retaining mechanisms only when all three losses fell below $0.1$. If no such mechanism was found, we decreased $\delta$; if one was found, we increased $\delta$, and repeated this process until the binary search interval was at most $0.1$.

## D.6 OTHER DETAILS

**Programming Languages, Packages, and Licensing** We ran our experiments in Python 3.7 (Van Rossum & Drake Jr, 1995), using NumPy (Harris et al., 2020), Jax (Bradbury et al., 2018), Haiku (Hennigan et al., 2020), and JaxOPT (Blondel et al., 2021). All figures were graphed using Matplotlib (Hunter, 2007).

Python software and documentation are licensed under the PSF License Agreement. Numpy is distributed under a liberal BSD license. Pandas is distributed under a new BSD license. Matplotlib only uses BSD compatible code, and its license is based on the PSF license. CVXPY is licensed under an APACHE license.

**Computational Resources** The experiments were conducted using Google Colab, which provides cloud-based computational resources. Specifically, we utilized an NVIDIA T4 GPU with the following specifications: GPU: NVIDIA T4 (16GB GDDR6), CPU: Intel Xeon (2 vCPUs), RAM: 12GB, Storage: Colab-provided ephemeral storage.

**Code Repository** the full details of our experiments, including hyperparameter search, final experiment configurations, and visualization code, can be found in our code repository (https://anonymous.4open.science/r/Dynamic-Mechanism-Design-ICLR-2026–081E).