# OpenReview forum: "Mechanism Design as Inverse Games: A Computational Approach to Dynamic Mechanism Design"
_ICLR.cc/2026/Conference — Submitted to ICLR 2026_

### Official Review · Reviewer_xLVM · 2025-10-23

**Soundness:** 2
**Presentation:** 2
**Contribution:** 2
**Rating:** 4
**Confidence:** 4

**Summary:**

The paper formulates mechanism design in dynamic environments as an inverse game problem within a partially observable Markov game (POMG) framework. The mechanism designer first sets the game parameters, after which agents play to reach equilibrium—analogous to a two-stage Stackelberg game.

The desired mechanism must satisfy incentive compatibility (IC) and individual rationality (IR) constraints. To find such mechanisms, the authors reformulate the problem as a constrained min–max optimization, where the designer seeks parameters that maximize the principal’s objective while ensuring IC and IR.

They propose two computational methods to handle partial observability: (1) Bayesian belief-state tracking for discrete type spaces, and (2) recurrent neural embeddings for continuous types.

Experiments in bandit auction settings show that their approach recovers known optimal single-item mechanisms and discovers new incentive-compatible mechanisms in complex multi-item environments lacking analytical solutions.

**Strengths:**

1. Provides a general computational framework for dynamic mechanism design, applicable to both discrete and continuous types as well as multi-item settings.
2. Introduces a novel min–max optimization approach that enforces IC and IR constraints without relying on Lagrangian method like previouse works in differentiable economics.

**Weaknesses:**

1. Lacks sufficient justification for formulating the problem as a min–max optimization in Equation (4) and for not adopting the traditional Lagrangian approach.
2. Despite proposing a general framework, experiments are limited to bandit auction settings, restricting the empirical validation.
3. Does not provide a comparison with versions that use Lagrangian relaxation, making it unclear how much improvement the proposed method offers.

**Questions:**

1. Why is the min–max formulation in Equation (4) necessary, and why not use the traditional Lagrangian method?
2. How does the proposed approach compare with versions that use Lagrangian relaxation in terms of performance or stability?

---

### Official Review · Reviewer_2BSW · 2025-10-30

**Soundness:** 3
**Presentation:** 2
**Contribution:** 3
**Rating:** 4
**Confidence:** 3

**Summary:**

The paper proposes a computational framework for dynamic mechanism design (DMD) which explicitly casts the problem as an inverse game theory one. Mechanisms are parameterized and mapped to induced partially observable Markov games (POMGs). The incentive compatibility constraint (IC) is enforced by forcing the exploitability of the truthful reporting policy to zero, and individual rationality constraint (IR) is enforced as non-negativity of agent payoffs. The overall objective is described as a min–max problem with a binary search over a target principal payoff, solved by two-timescale SGDA with either (i) explicit Bayesian belief tracking (for discrete types) or (ii) recurrent neural embeddings (for continuous types). Theoretical results (Theorem 4.1) give polynomial-time convergence under structural assumptions. Experiments on bandit auctions recover single-item benchmarks and produce high-payoff mechanisms in multi-item settings.

**Strengths:**

Overall, I find the direction taken by the paper to be very interesting. Results appear to be technically sound. The experimental evaluation is promising and shows that the proposed technique recovers closed form solutions in simple settings, while being able to generalise to more complex ones.

**Weaknesses:**

Compression maps are assumed to exist without much justification. It would be interesting to understand when this is a reasonable assumption, and when it’s not. I guess there may be many settings where this is non-trivial to verify. Moreover, I really don’t like the fact that formal assumptions 1--4 are only stated in the appendix, this does not help in clearly understanding subsequent results. Without any discussion it’s unclear whether they hold in common DMD applications. This should be discussed in the main paper.

It would be nice to have a more detailed discussion around the reasons for which the naive approach cannot be applied (paragraph at line 252), and to provide better evidence for this.

In some places I would appreciate if the authors could be more precise in defining concepts before referring to them (e.g., global vs local constraints (paragraph at line 172); discrete-type compressed information setting (Line 275)).

Other comments: Notation is sometime very cluttered (eg constraint 2 in the LP at page 5). It is quite annoying that the references are only located in the supplementary file, which is a separate one from the main paper.

**Questions:**

1) Is assumption on type evolution at line 151-152 standard in the DMD literature?

2) Could the authors provide other concrete examples where the assumptions are satisfied?

---

### Official Review · Reviewer_4jby · 2025-10-31

**Soundness:** 3
**Presentation:** 3
**Contribution:** 2
**Rating:** 4
**Confidence:** 4

**Summary:**

The paper provides a computational framework for solving optimal dynamic mechanism design problems. It is based on a min-max formulation that encodes the incentive compatibility constraint. To handle partial observability, it proposes a Bayesian belief-state tracking process that has convergence guarantees. To tackle the more challenging where agents have continuous types, it proposes using recurrent neural embeddings. The effectiveness of the framework is shown through its ability to discover dynamic mechanisms in new settings.

**Strengths:**

The paper proposes a sound computational framework for dealing with an interesting and well-studied problem, dynamic mechanism design. The formulation is quite natural and comes with provable guarantees. Furthermore, the reformulation as a min-max problem makes it possible to leverage existing techniques to solve such problems. In other words, the paper provides a reduction from dynamic mechanism design in partially observable settings to min-max optimization. From a broader perspective, the paper contributes to an active line of work, called differentiable economics, that leverages the power of neural networks to address problems in economics where analytic solutions have not been forthcoming. The experiments show that the proposed framework is able to i) recover existing mechanisms that were derived analytically, and ii) find new mechanisms for more challenging settings. Furthermore, the writing and presentation of the paper is very good.

**Weaknesses:**

On the negative side, the paper can do more to clarify certain connections with prior work. For example, it cites a paper by Goktas et al. (2024) for solving the inverse equilibrium problem; there were earlier references that could be included here, for example, "Inverse Game Theory" by Kuleshov and Schrijvers. I would also suggest commenting further on the technical similarities between this present paper and the paper of Goktas et al. (2024); from my reading, there are several similarities, and the present paper should explain more clearly the parts of the formulation that are new. Another closely related paper is "Computing Optimal Equilibria and Mechanisms via Learning in Zero-Sum Extensive-form Games" by Zhang et al. (2023). That Lagrangian formulation of that paper is closely related to Theorem 3.1 and the subsequent Algorithm 1, which is also based on performing a certain binary-search procedure. The paper by Zhang et al. also addresses sequential auction design using deep learning, so the authors should discuss similarities and differences with that paper.

From a theoretical standpoint, I am struggling to fully understand the key assumptions that are necessary to make the approach sound. Partially observable MDPs are notoriously hard, but the paper makes some further assumptions to obtain Theorem 4.1. Some of those assumptions appear to be overly restrictive, such as Assumption 4---that transitions are independent of outcomes. From a theoretical standpoint, relying on partially obserable MDPs instead of extensive-form makes the problem computationally much harder, which is why such artificial assumptions are needed. Can the authors comment more on those assumptions and why they believe they are justified in practice?

Finally, there are some parts of the presentation that are sloppy. Figure 2, for example, doesn't properly show the x-axis and there is an overlap between different labels that makes it hard to parse.

**Questions:**

See above.

---

### Official Review · Reviewer_Nimc · 2025-10-31

**Soundness:** 2
**Presentation:** 1
**Contribution:** 2
**Rating:** 2
**Confidence:** 3

**Summary:**

This paper proposes a computational framework for dynamic mechanism design (DMD) under Bayesian Nash incentive compatibility by formulating the problem as an optimization over parameterized games. The authors generalize the inverse equilibrium framework—previously applied to static games—to the dynamic setting. Their approach introduces constraints for truthfulness and individual rationality, treats dynamic mechanisms as parameterized partially observable Markov games (POMGs), and develops gradient-based optimization methods to search for equilibria consistent with truthful reporting.
In special cases, the authors show that their method recovers known classical results, while also demonstrating empirical performance in more complex multidimensional auction-like environments where analytical solutions are unavailable. They also propose a gradient-dominance-based convergence condition, though key details are relegated to the appendix rather than the main text.

**Strengths:**

The paper’s conceptual ambition is commendable: connecting dynamic mechanism design, inverse game theory, and differentiable optimization is a meaningful direction.
It represents an ambitious and timely attempt to unify inverse game theory and dynamic mechanism design, and is a potentially valuable computational approach to problems with evolving types and partial observability.
Experimental results appear to replicate known benchmarks and extend to new, analytically intractable domains.
The overall research direction (using differentiable optimization tools for DMD) is promising.

**Weaknesses:**

As written, the paper suffers from significant problems of notation, exposition, and logical clarity. These severely obscure the contribution, and at least one major technical component—the formulation and interpretation of the main optimization problem and Theorem 4.1—appears inconsistent or incorrect.

The paper suffers from severe notational overload and redundancy. The paper repeatedly introduces variations or re-explains concepts and symbols without clear motivation.


Presentation. The paper’s language is often verbose and repetitive, masking rather than clarifying the conceptual advances. This is particularly problematic, as key details are relegated to the appendix, and novel contributions go severely underexplained, while the basic model setup is severely overexplained (Primarily: the components of the game, and their notation, are defined and explained, and then redefined and re-explained in the context of truthfulness and incentive compatibility).

Detailed feedback:


[Lines 101–120]
 The paper defines each model component (transition kernel, reward function, observation process) sequentially before showing how they interact. Integrating definitions with their use (e.g., in the POMG tuple) would make the exposition far clearer. As written, new symbols accumulate before the reader understands their purpose.

[Line 114]
 The independence of reward functions is not stated or justified. If independence is assumed, please explain whether this assumption affects only convergence rates or whether it constrains generality. If dependence exists, show how it is handled mathematically.

[Lines 114–116]
 Only the transition and reward functions are parameterized by Θ, while the observation kernel is not. This asymmetry seems arbitrary. If the observation process depends on Θ, it should be parameterized consistently. Otherwise, explain why this simplification is valid, or why not doing so would cause problems in the analysis.

[Lines 114–120]
The notation τ ∈ [(T−1)*] is nonstandard—please define it explicitly.


The symbol d is used ambiguously (is it a variable or differential operator?). If differential, it should probably be typeset upright.


The variable τ appears as both subscript and superscript in places—choose one convention for time indices.



[Line 132] Crucial elements of the Bayesian formulation are deferred to the appendix. Without this material, the main text gives an incomplete picture of the equilibrium concept (it's just a regular NE), and belief states become important much later in the paper.

[Line 362]  Similarly, convergence proofs and assumptions are entirely left for the appendix. These are essential for evaluating the soundness of the proposed optimization method.

[Lines 135–140] The notation induces confusion between stochastic and deterministic policies.
 The interchange between P, π, and $P^{PO}$ is confusing. It is unclear whether the analysis applies to pure, mixed, or stochastic Nash strategies. Please formalize this consistently. I think the standard mixed definition, or the standard pure definition, would both suffice, as you only consider a single pure equilibrium anyway.


[Lines 101–210]
 Sections 3.1–3.2 repeat *essentially* the same definitions three or four times (once for general actions, again for types, then for reporting). The symmetry between transition kernels and type evolution, and between observation kernels and observation of types, could be unified in one coherent definition instead of restating the structure each time.

[Line 219]
 Please state explicitly and early on that the paper deals with Bayesian-Nash incentive compatibility (BNIC) rather than dominant-strategy IC (DSIC): this limitation was ambiguous.


[Line 132]
 Crucial elements of the Bayesian formulation of BNE are deferred to the appendix. Without this material, the main text gives an incomplete picture of the equilibrium concept.

[Line 362]
Similarly, convergence proofs and assumptions are entirely left for the appendix. These are essential for evaluating the soundness of the proposed optimization method.

[Overall] The repetitive presentation and unnecessary introductions distract from the paper content, and seem to be the reason why discussion of the actual contributions was so sparse.   Very important details are given only in the appendix, and if the paper were not so repetitive, they could be made explicit and explained, both on a technical and an intuitive level.

**Questions:**

Why was measure theoretic notation and definition even necessary? It seems to obscure a relatively simple stochastic process definition, and I didn't see much in the way of a payoff for doing so.

Line 220: You defined  ε-optimality earlier, in the sense of ε-equilibria, so why not  define the DMD setting in terms of that?


[Theorem 4.1, Lines 250–290]
 The theorem as written is not clear:
α and β > 0 are introduced as constants, but are used inside a min–max expression without quantifiers.


Is the result meant to hold for all α, β, or for some fixed choice?


Moreover, for any value of α, β > 0, the only feasible θ would make all agent utilities 0, collapsing the optimization problem to consider only games where all agents can only just barely rationalize play. This can’t be what was intended, please clarify.

---

### Meta-Review · Area_Chair_VaGR · 2026-01-05

**Summary:**

This paper on mechanism design received four negative reviews, and the reviewers questions and concerns were not answered.

Clear reject

**Reviewer Scores:**

N/A

---

### Decision · Program_Chairs · 2026-01-26

Reject